# *β*-Variational autoencoders and transformers for reduced-order modelling of fluid flows

Alberto Solera-Rico [1,2], Carlos Sanmiguel Vila [1,2], Miguel Gómez-López [2], Yuning Wang [3], Abdulrahman Almashjary[4], Scott T. M. Dawson [4] & Ricardo Vinuesa [3] ✉

Variational autoencoder architectures have the potential to develop reduced-order models for chaotic fluid flows. We propose a method for learning compact and near-orthogonal reduced-order models using a combination of a *β*-variational autoencoder and a transformer, tested on numerical data from a two-dimensional viscous flow in both periodic and chaotic regimes. The *β*-variational autoencoder is trained to learn a compact latent representation of the flow velocity, and the transformer is trained to predict the temporal dynamics in latent-space. Using the *β*-variational autoencoder to learn disentangled representations in latent-space, we obtain a more interpretable flow model with features that resemble those observed in the proper orthogonal decomposition, but with a more efficient representation. Using Poincaré maps, the results show that our method can capture the underlying dynamics of the flow outperforming other prediction models. The proposed method has potential applications in other fields such as weather forecasting, structural dynamics or biomedical engineering.

Turbulent flows are an important and ubiquitous phenomenon in nature and engineering, with applications ranging from aircraft design to weather forecasting. Understanding the behaviour of fluid flows is often challenging due to their complex spatio-temporal dynamics involving a large number of degrees of freedom and complex non-linear interactions[1]. As a result, there is a growing interest in developing reduced-order models (ROMs) of fluid flow dynamics that can capture the key underlying dynamics of the flow while reducing the problem dimensionality[2,3]. Developing ROMs is one of the most prominent research fields since they facilitate finding low-dimensional representations that can be applied to perform flow control applications[4] or reduce the computational cost of numerical simulations[5,6]. One of the most used techniques for dimensionality reduction in the fluid-dynamics community is proper orthogonal decomposition (POD), which involves finding the dominant modes of variation in a given dataset and projecting the data onto a lower-dimensional subspace spanned by these modes. Another famous linear approach is the dynamic-mode decomposition (DMD), which identifies dynamic modes that govern the evolution of the system over time[7]. While POD and DMD, as well as the extensions of these methods such as the spectral POD[8] or the higher-order DMD[9], have successfully reduced the dimensionality of some flows[10], their optimal linear bases exhibit limitations when working with turbulent flows, which typically involve complex non-linear interactions[11].

In recent years, machine-learning (ML) techniques have emerged as promising approaches for developing ROMs of fluid-flow dynamics[6,12–17]. One of the potential ML techniques adopted to create non-linear ROMs is the neural networks with convolutional

[1]Aerospace Engineering Research Group, Universidad Carlos III de Madrid, Leganés, Spain. [2]Subdirectorate General of Terrestrial Systems, Spanish National Institute for Aerospace Technology (INTA), San Martín de la Vega, Spain. [3]FLOW, Engineering Mechanics, KTH Royal Institute of Technology, SE-100 44 Stockholm, Sweden. [4]Mechanical, Materials, and Aerospace Engineering Department, Illinois Institute of Technology, Chicago, IL 60616, USA. ✉e-mail: rvinuesa@mech.kth.se

autoencoder architectures[18,19]. These architectures comprise of both an encoder and a decoder trained to minimise the reconstruction error between the encoded-decoded data and the initial data. The resulting encoder allows obtaining a latent-space composed of non-linear representations. Then, neural-network architectures suitable for temporal predictions[12,20,21], such as long short-term memory (LSTM) networks, can be used to model the dynamics of the non-linear latent-space, resulting in a fast surrogate model for fluid-flow predictions.

Among the different autoencoder architectures, variational autoencoders (VAEs) have proven to be effective for encoding the spatial information of fluid flows in non-linear low-dimensional latent-spaces[12,22]. Unlike a standard autoencoder, a VAE architecture is based on a probabilistic framework for describing an observation in latent-space by including an additional loss term on the latent-space variables. However, the low-dimensional representations obtained using these architectures lack the orthogonality of the classical linear-decomposition techniques. To overcome this issue, the $\beta$-VAE architecture introduced in Refs. 23,24 modifies the loss function of the VAE by adding a regularisation parameter to balance the reconstruction accuracy with regularisation and latent-space disentanglement. The value of the parameter $\beta$ is chosen high enough to produce a near-orthogonal latent-space representation but as small as possible to avoid increasing the reconstruction error. The potential of these architectures to develop a compact and near-orthogonal ROM was reported in Ref. 25, where they tested this architecture in a high-fidelity simulation of a turbulent flow through a simplified urban environment. The results showed a ROM that was able to capture up to 87.36% of the original energy with only five variables in the latent-space, compared to the 32.41% obtained with five POD modes.

For the temporal predictions, the LSTM has been shown to be an effective architecture in turbulent flows[20,26,27]. However, in the latest years, another neural-network architecture known as transformer[28] appears to have the potential to outperform the LSTM and allow the development of more complex ROMs. The transformer is a deep neural-network architecture that has gained widespread attention in recent years due to their state-of-the-art performance in natural-language-processing (NLP) tasks such as language translation[29] or text generation[30]. Unlike traditional recurrent neural networks like LSTM, which process sequential data one element at a time, transformers are designed to capture long-range dependencies between elements in a sequence. This capability is achieved using attention mechanisms which allow the network to attend to different parts of the input sequence at each network layer. As a result, transformers have been shown to outperform previous state-of-the-art methods in several NLP benchmarks, often by large margins.

The success of transformers in NLP has led to their application in other domains, including computer vision[31,32], audio processing[33], and robotics[34]. In particular, due to their ability to capture long-range dependencies, transformers are particularly well suited to model dynamic systems[35]. In fact, transformers are able to represent the multi-scale character of turbulence in long temporal sequences[36]; this can only be captured by LSTMs when separately predicting modes of different ranges of frequencies[37]. In these applications, the goal is to learn a low-dimensional representation of the system that captures the underlying dynamics, which can then be used to make predictions or generate new trajectories. Transformers are a promising tool for this task, as they can learn complex temporal dependencies and capture long-term trends in the data while allowing efficient parallel processing.

The potential combination of autoencoder architectures, which enable obtaining near-orthogonal non-linear latent-spaces, with transformer architectures for the dynamics of the temporal predictions, is a powerful tool that can be employed to model complex flows with a higher level of accuracy. For this reason, in this paper, we propose a $\beta$-VAE and transformer-based model for encoding the fluid-flow velocity fields and learning a ROM of its spatio-temporal dynamics. Two flow cases are analysed, namely a periodic and a chaotic configuration of a two-dimensional, viscous flow over two collinear flat plates, obtained by numerical simulation. Flow past multiple bodies in close configuration is relevant for a range of applications, such as buildings or chimneys in urban and industrial environments, power lines, offshore structures, and heat exchangers. Even at relatively low Reynolds numbers, such flows can exhibit substantially more complexity than flow over a single, isolated body[38–40].

The resulting latent-space of $\beta$-VAEs is analysed using POD as a reference case to analyse the resulting spatial mode features, and different latent-spaces are tested. The temporal predictions of the latent-space dynamics performed using transformer-based architectures are compared with other ML temporal models, including LSTMs and Koopman with Non-linear Forcing[41]. Finally, the predictions are assessed using the reconstructed predicted fields and Poincaré maps to assess the dynamic behaviour of the resulting ROMs.

## Results
### Analysis of the latent-spaces
In this section, the $\beta$-VAE architecture and the POD are applied to two flow cases generated from a numerical simulation of a two-dimensional, viscous flow around two collinear flat plates. The characteristics of both cases are discussed next:
- Periodic flow with Reynolds number based on the freestream velocity $U_\infty$ and (single-plate) chord length $c$ of $Re = 40$, where the two collinear plates are arranged at an angle of 90° with respect to the incoming flow. Total of 1000 instantaneous flow fields.
- Chaotic flow with Reynolds number of $Re = 100$, where the two collinear plates are arranged at an angle of 80° with respect to the incoming flow. Total of 150,000 instantaneous flow fields.

In both cases the domain where data is collected has dimensions $96c \times 28c$, where the spatial resolution is $300 \times 98$ grid points with a uniform grid spacing. The separation between snapshots is one convective time $\Delta t = c/U_\infty = t_c$ for periodic case and $\Delta t = c/U_\infty/5 = t_c/5$ for chaotic case (downsampled in both space and time from the original simulations). A more detailed dataset description and a sketch of the flow configuration is given in the Methods section.

To compare the performance of $\beta$-VAE and POD as defined in Methods, we define, following Ref. 25, the energy percentage $E$ that is captured by the low-order reconstruction as:

$$E = \left(1 - \left\langle \frac{\sum_{}^{N_p} \sum_{i=1}^{2} (u_i - \tilde{u}_i)^2}{\sum_{}^{N_p} \sum_{i=1}^{2} u_i^2} \right\rangle \right) \times 100\%, \tag{1}$$

where $\langle \cdot \rangle$ indicates ensemble averaging in time, $N_p$ is the number grid points along the spatial domain $u_i$ denotes the $i^{th}$ reference value of fluctuating component velocity and $\tilde{u}_i$ its low-order reconstruction, respectively.

To assess the orthogonality of the latent variables, we calculate the correlation matrix $\mathbf{R} = (\mathbf{R})_{dxd}$, which is defined as follows:

$$R_{ii} = 1, \quad R_{ij} = \frac{C_{ij}}{\sqrt{C_{ii} C_{jj}}}, \tag{2}$$

for all $1 \le i \ne j \le d$ where $C_{ij}$ denotes the components $i, j$ of the covariance matrix $\mathbf{C}$ and $d$ is the dimension of the latent-space. A value of 0 is reached when all the variables are completely uncorrelated ($R_{ij} = 0$) and one when they are completely correlated ($R_{ij} = 1$). This metric reports the degree of correlation between the latent variables.

The modes have been ordered according to their cumulative contribution to the reconstructed energy, $E$, following Ref. 25. The first mode is chosen as the one with the largest individual contribution to $E$, and the next are those that have the maximum contribution when added to the previous modes.

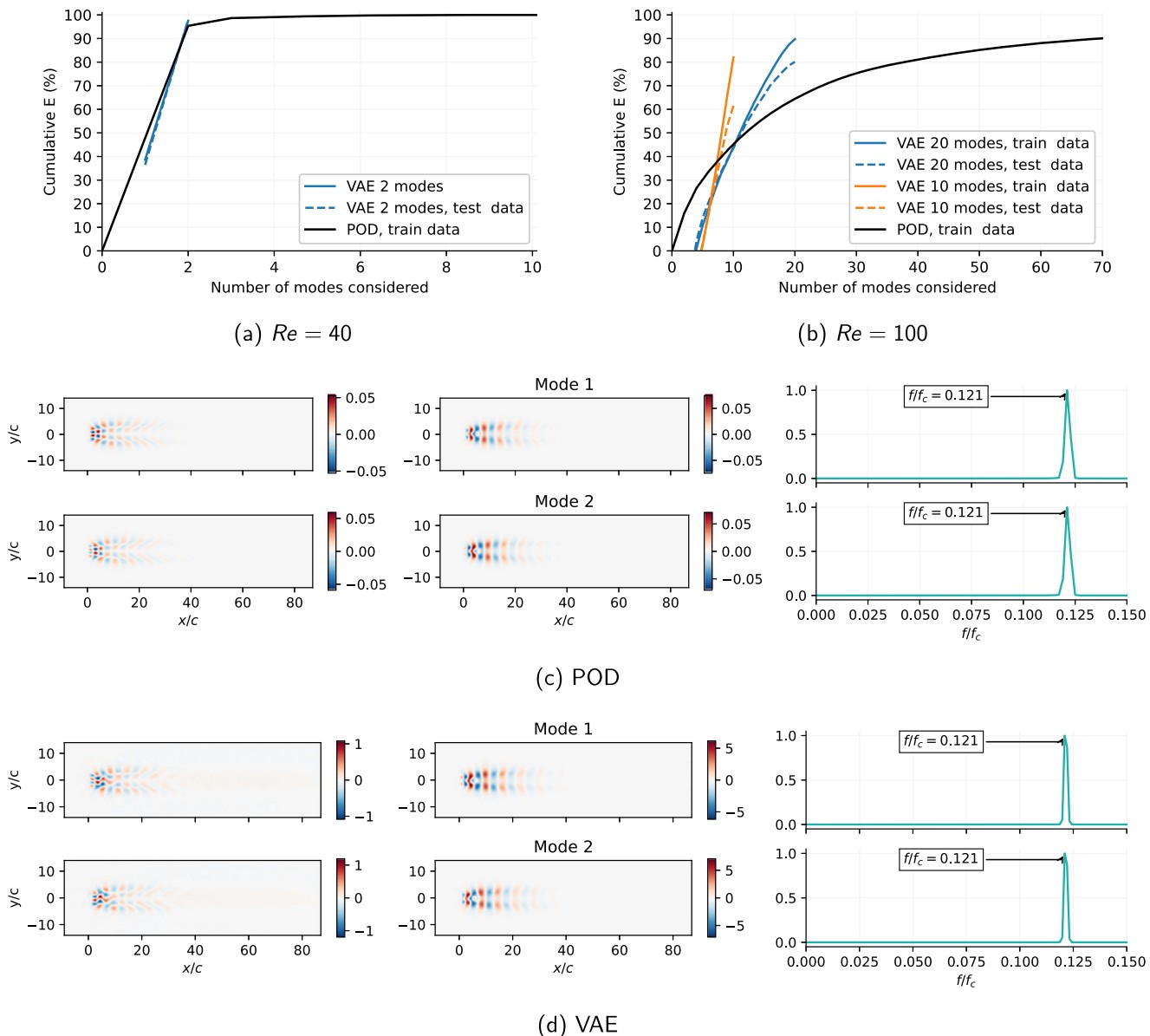

**Fig. 1 | Proof of concept analysis with periodic flow case. a, b** Fraction of energy reconstructed by POD and $\beta$-VAE as a function of number of modes. **a** $Re = 40$, $\alpha = 90°$, (**b**) $Re = 100$, $\alpha = 80°$. **c, d** Resulting modes for the $Re = 40$, $\alpha = 90$ case: (**c**) POD, (**d**) $\beta$-VAE. The first column contains the streamwise-velocity component modes sampled with a unit value, and the second one the crosswise-velocity component. The third column represents the frequency content in the temporal coefficient associated with each of the modes ($f_c = U_\infty/c$); note that in (**d**) this frequency content is evaluated in the latent-space.

The periodic case is used as a benchmark to validate the physical soundness of the $\beta$-VAE model representations. This flow case can be adequately represented only using two POD modes that can represent $E = 98.4\%$ of the kinetic energy of the fluctuations with respect to the base flow, as observed in Fig. 1a. Figure 1c represents the two most energetic POD modes, which are identified as a shedding wake. For the $\beta$-VAE case with $\beta = 0.001$, the results in Fig. 1d show the spatial modes. The $\beta$-VAE spatial modes are defined as the result of using the $\beta$-VAE decoder network with an input vector, $s_i$, containing a unit value in the desired $i^{th}$ element and zero elsewhere, $s_i = \delta_i^j = (0, \cdots, 1, \cdots, 0)$, that gives as a result the corresponding spatial $i^{th}$ mode. For a detailed description, the reader is referred to the Methods section. The reconstructed energy in this case is equal to $E = 97.5\%$, with a cross correlation coefficient, $R_{12} = 0.0015$. It is observed from Fig. 1c, d that the spatial modes obtained from both methods exhibit the same pattern of shedding flow. The spectrum of the temporal coefficients for both methods is shown in the last column of Fig. 1c, d. The spectra

analysis is from the resulting $r_i(t)$ and $a_i(t)$ time coefficients for the $\beta$-VAE and POD, respectively. This further confirms that the dynamics associated with the spatial modes are also in good agreement: both methods can capture the same characteristic frequency. This result shows that the latent-space also exhibits meaningful physical phenomena of the flows.

Since the $\beta$-VAE architecture requires the user to set a latent-space dimension $d$, it can be argued that $d$ can be set to a value larger than 2. However, we observed that models with larger latent-spaces produce only two meaningful modes, as the remaining modes have negligible values. This behaviour shows that the $\beta$-VAE regularisation effectively avoids the artificial creation of more modes than necessary to represent the solution. In the work by Eivazi et al.[25] it was shown that the $\beta$-VAE produces compact representations of latent-spaces, which suggests that these architectures may be a good framework in cases that can be represented with few energetic phenomena.

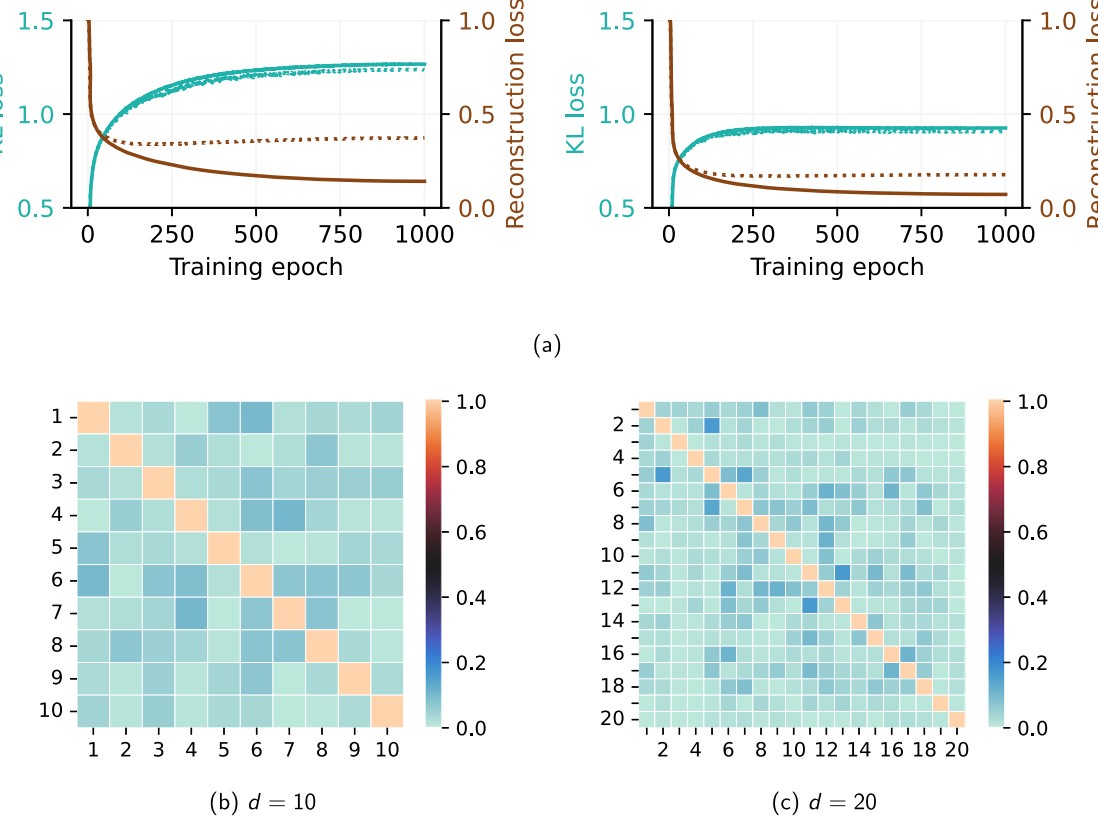

**Fig. 2 | Training and latent space analysis. a** Evolution of $\beta$-VAE losses during training: (left) 10 modes, (right) 20 modes, (solid) train, (dashed) test. Losses are defined in the Methods section. **b, c** Correlation matrices corresponding to the $\beta$-VAE mode coefficients for the case with $Re = 100$, $\alpha = 80°$.

After this first assessment of the latent-spaces created by the $\beta$-VAE architectures, a non-linear and higher-dimensional chaotic case at $Re = 100$ with the two collinear plates arranged at an angle of 80° with respect to the incoming flow is tested. As well as the higher Reynolds number, the change in angle adds additional complexity to the configuration as the geometry is no longer symmetric with respect to the freestream. In this case, the size of the latent-space is set to be as small as possible but large enough to allow the separation of different physical effects in the resulting modes. In this relatively complex case, it is expected to find more variety of effects in the fluid flow, requiring a more nuanced mapping to the low-dimensional latent-space, and an evident lack of performance of linear methods such as POD.

To define an appropriate $\beta$ value, different values were tested between 0.001 and 0.4 and the correlation matrix and $E$ were calculated to evaluate the performance in both metrics with a fixed latent-space. As previously mentioned, the value of $\beta$ is chosen high enough to produce a near-orthogonal latent-space representation but as small as possible to avoid increasing the reconstruction error, increasing $\beta$ has also been found to slightly improve the generalisation of the model, balancing train and test metrics. After this study, a $\beta = 0.05$ is chosen, and an analysis of the appropriate latent-space is followed. For this analysis, the loss terms from the $\beta$-VAE are analysed during the training process as reported in Fig. 2a, with a particular interest in the Kullback-Leibler (KL) divergence loss. The figure shows that, as expected, the reconstruction loss is lower for the $\beta$-VAE model with larger latent-space $d = 20$, as more information is allowed to flow through the autoencoder bottleneck. The less constrained latent-space also allows for a lower KL divergence loss, meaning the latent-space distributions are closer to standard normal distributions. This lower KL loss also reflects the better ability of the model to produce disentangled representations in the latent-space. In this case, $E = 89.8\%$ for train data and $E = 80.1\%$ for test data,

as seen in Fig. 1b, indicating a good reconstruction and generalisation capability.

On the other hand, the model with a smaller latent-space $d = 10$ converges to a relatively higher reconstruction loss. Even the KL loss term takes longer to converge during the training process because the bottleneck is too strict. For $d = 10$, the reconstructed energy $E = 82.0\%$ for train data and $E = 61.6\%$ for test data, reflect the loss of reconstruction capability due to the constrained bottleneck. The information is compressed into a few modes with less freedom for disentangled representation, a fact that implies that even for these architectures, a minimum number of modes is required to obtain an appropriate latent-space. It was also observed that the $\beta$-VAE architecture tends to overfit the training data if the latent-space is insufficient to achieve proper disentanglement. Although the model with a latent-space of size $d = 10$ is considered valid, the loss of generalisation, as observed in Fig. 2a with a poor performance in the test data, motivates the choice of $d = 20$ for the model chosen as a reference for the present study. Apart from the latent-space dimension, the choice of $\beta$ is critical to ensure the generalisation of the latent-spaces. Lower $\beta$ values produce a large $E$ in train data but at the cost of much lower $E$ values in test data, which indicates overfitting. Increasing $\beta$ above a critical value also affects the performance by not only affecting the degree of disentanglement of the latent-space but also decreasing the maximum $E$ obtained, being the classical VAE architectures ($\beta = 1$) tested with a poor performance in both train and test data. Apart from that, we have also tested a vanilla autoencoder that was unable to generalise the representation and did not produce a disentangled latent-space, which reinforces the ability of the $\beta$-VAE to perform a better generalisation[23].

Figure. 3 shows, for the first 6 modes, the $u$ and $v$ components and the spectrum obtained form the temporal modes associated with each mode, while Fig. 4 shows the same quantities obtained using POD. The

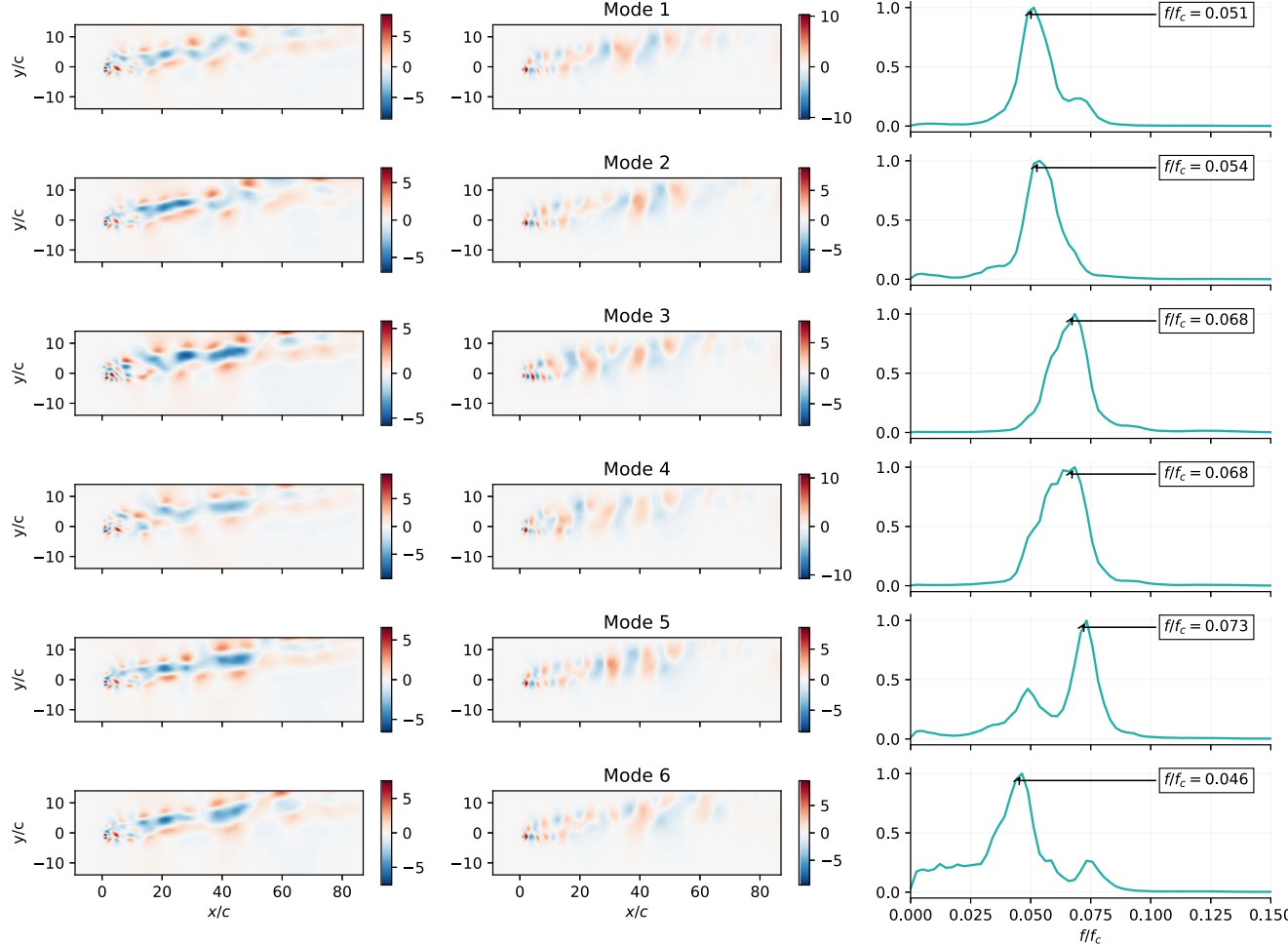

**Fig. 3 | Analysis of $\beta$-VAE modes.** Resulting $\beta$-VAE six first spatial modes for $Re = 100$, $\alpha = 80°$ ranked according to their contribution to $E$. The first column contains $u$ values for modes sampled with $s_i = 1$. The second column shows the $v$ values for the same inputs. The third column represents the frequency content of each individual mode ($f_c = U_\infty/c$).

spectral analysis of the temporal dynamics of the modes is used to identify different modes representing the same physical effect and provides helpful information for determining the size of the latent-space, since it allows more meaningful comparison with the POD. In this case, the $\beta$-VAE spatial modes are a non-linear combination of the most relevant flow features. Comparing the $\beta$-VAE and POD spectra, it can be seen that the $\beta$-VAE model can better separate different phenomena, each of which has its characteristic frequency associated with it, which appears as a peak in the spectrum. The different effects can be further isolated as individual modes by using larger latent-spaces. However, it is observed that the frequencies associated with the most energetic POD modes appear in the $\beta$-VAE dynamics, suggesting the idea that the $\beta$-VAE latent-spaces can find non-linear modes that effectively represent the dynamics of the system. The effect of the mode disentanglement produced by $\beta$-VAE models can be observed in the correlation matrix between the time series of each mode, Fig. 2b, c. As reported, the correlation between different modes of the time series is almost negligible due to the near-orthogonal representation in the latent-space. It is observed that the $\beta$-VAE also appears to be able to find pairs of modes that represent the orthogonal components of the harmonics of the vortex-shedding process that represents the large-scale convective structures of the vortex wake studied[42], such as in POD modes.

### Latent-space predictor models

In this section, the temporal dynamics of the latent-space $\mathbf{r}(t)$ for the case $d = 10$ and $d = 20$ are combined with different ML models to

implement a framework able to predict the temporal dynamics in the latent-space that can be used with the decoder from the $\beta$-VAE to obtain flow-field temporal predictions. With this purpose, two transformer models, self-attention[28] and easy attention[43], a Koopman with Non-linear Forcing (KNF) model[41], and an LSTM network[44] are implemented and compared. The KNF and LSTM models have previously been analysed and compared in Ref. 27 and applied to predict the temporal dynamics of a low-order model of near-wall turbulence, showing that both approaches can reproduce the temporal dynamics of this system. Furthermore, the transformer has been used in the context of temporal predictions of turbulent flows in Ref. 36. The four model architectures were tuned to obtain the lowest mean-squared error over the validation data, with the self-attention transformer later discarded for clarity due to the significantly better results obtained by the easy-attention model.

All three predictor models are trained to predict the next time step of a temporal sequence of previous latent-space $r(t)$ vectors. Further details of the process and hyperparameter choice are detailed in the Methods section. Once trained, the models are used recursively to predict the next time steps. By predicting long time series, we can determine whether the model has been able to learn the system dynamics correctly. Fig. 5 shows the reference values $r_i(t)$ obtained by encoding the test data using the previously trained $\beta$-VAE and the corresponding $p_i(t)$ predictions from each model.

As expected for a chaotic dynamical system, all predictor models diverge from the original trajectory after several time steps and appear

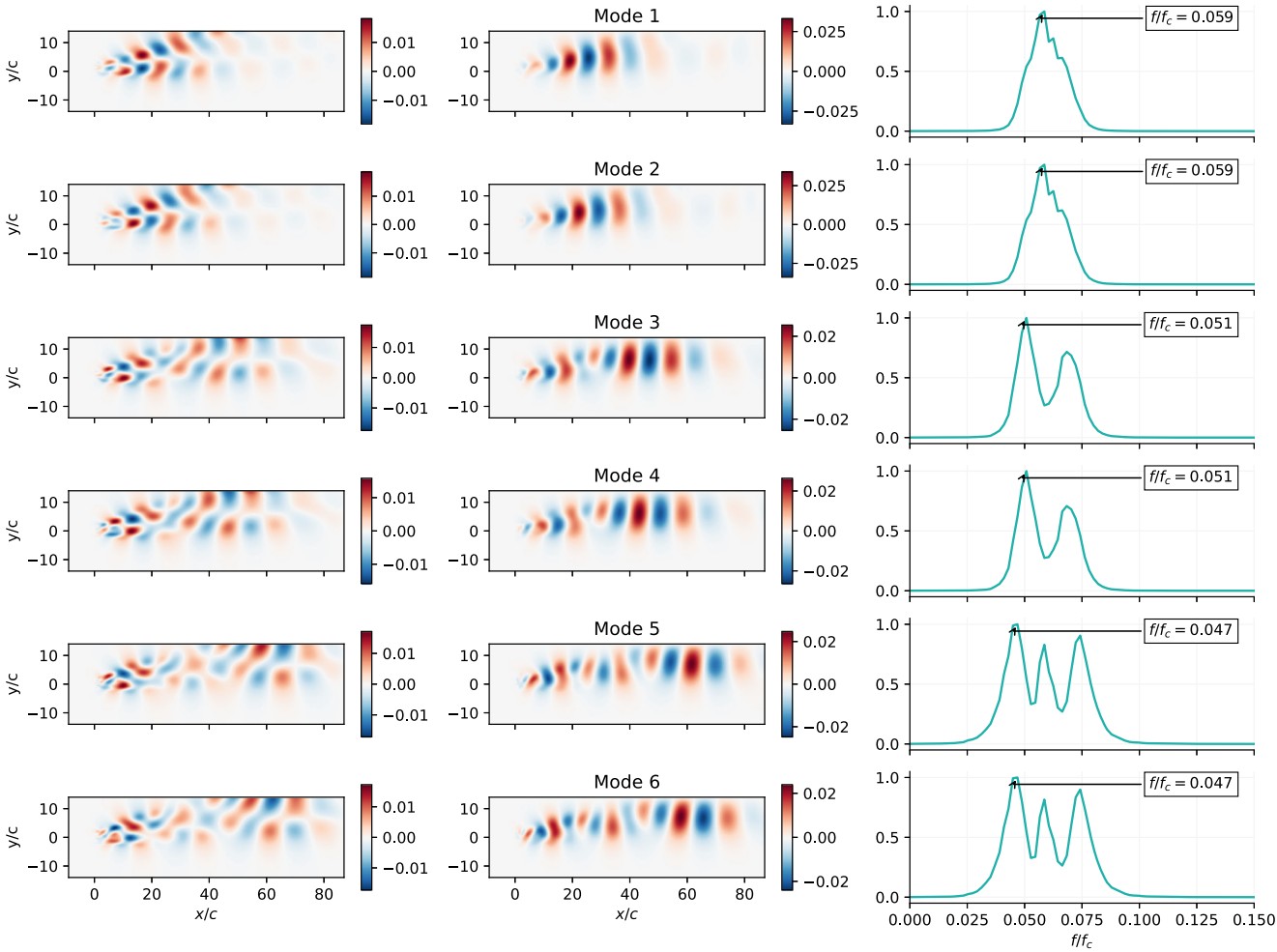

**Fig. 4 | Analysis of POD modes.** POD spatial modes for $Re = 100$, $\alpha = 80°$. The first column contains $u$ values for each POD mode and the second column shows the $v$ values. The third column represents the frequency content of each individual mode $a_i(t)$ ($f_c = U_\infty/c$).

to exhibit similar quantitative performance. To further evaluate the deviation error after several time steps, the ensemble average of the L2 error norm to the prediction horizon is defined as:

$$\varepsilon(\Delta t) = \left\langle \left( \sum_{i=0}^{d} \left( p_i(t) - r_i(t) \right)^2 \right)^{1/2} \right\rangle_{\text{ensemble}}. \quad (3)$$

This quantity is shown in Fig. 6a and the ensemble is computed over 100 evenly spaced windows in the test data to capture the average trend. This plot shows how the error $\varepsilon(\Delta t)$ increases as the prediction horizon $\Delta t$ gets longer. It can be seen that the error growth of the easy-attention transformer is always one of the slowest, in particular for the models trained with $\Delta t = 0.2 t_c$. The better behaviour of the transformer for cases sampled more densely in time could be the result of a better ability to represent the multiple time scale phenomena present in the data. However, the error growth over the prediction horizon does not provide any information about the long-term behaviour of the predictions associated with proper learning of the system dynamics. Therefore, this aspect is further evaluated in terms of the dynamic behaviour of the system through the Poincaré-map analysis using the predictions for the 3000 test data time steps not seen during the training process.

The Poincaré map is constructed as the intersection of the latent vectors with the hyperplane $r_1 = 0$ on the $r_2 - r_3$ space with direction $dr_1/dt > 0$, using then a probability density function (PDF) to fit the resulting intersection points. These distributions are plotted for the true data and the model predictions in Fig. 6b–d. This figure shows the correlation between the amplitudes of the $r_2$ and $r_3$ latent variables at the intersection. It can be seen from these figures that the KNF models do not adequately reproduce the variability of this correlation. This result can be explained by the tendency of the KNF model to converge towards a harmonic behaviour as the prediction progresses, which does not capture the variability in the evolution of the temporal coefficients that represent this chaotic flow. The LSTM and, above all, the transformer, produce accurate predictions of the dynamics of the system using the patterns and correlations between the different $r_i(t)$ in the latent-space learned by the $\beta$-VAE. This comparison between models suggests that focusing only on instantaneous predictions may not be the correct approach to develop ROM models based on ML techniques. The choice of latent variables $r_i(t)$ for this figure is motivated by the fact that these are the most relevant in terms of reconstructing $E$. Still, this procedure was reproduced for all the combinations of $r_i(t)$, being the quality of the results of the Fig. 6b–d representative for all the cases. Finally, the first six probability density functions of the predicted latent-space are shown in Fig. 6e, where it can be seen that the transformer model is better able to capture the variability of the latent-space.

Using the latent vectors predicted with the transformer, we can develop a ROM of the flow in the latent-space that captures the underlying dynamics, as shown by the Poincaré maps, and then using these predicted $r_i(t)$ values the $\beta$-VAE decoder network can be used to

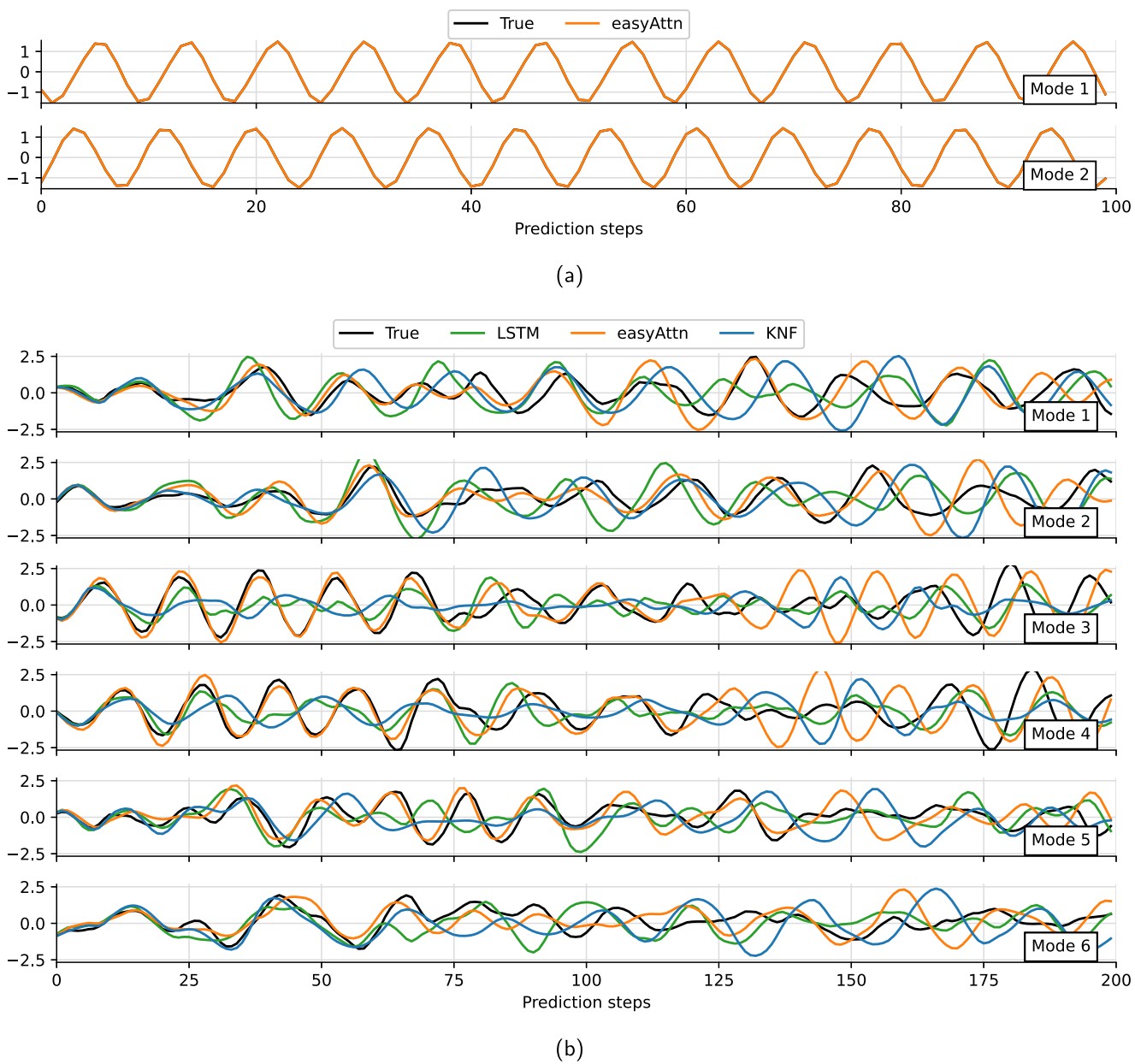

**Fig. 5 | Temporal evolution in the latent-space.** Example of a trajectory of the latent-space modes with their associated predictions by different models: (**a**) $Re = 40$, $\alpha = 90°$, (**b**) $Re = 100$, $\alpha = 80°$, six first modes.

project the predicted latent-space vector back into physical space, completing the flow field prediction model. The results of this prediction are shown in Fig. 7. In the figure, the first column shows the actual $t + n$ flow field in the dataset, the second column shows the same snapshot compressed into latent-space and decompressed by the $\beta$-VAE network, and the last column shows the field predicted by the ROM created using the $\beta$-VAE and each predictor model. To assess the performance of the complete ROM model, the reconstructed energy over the prediction horizon is calculated using an ensemble over 50 evenly-spaced windows in the test data. The results shown in Fig. 8a indicate a better reconstruction ability for models using $d = 20$, as expected due to the less restrictive autoencoder bottleneck. For the $d = 20$ case, models perform better when using $\Delta t = t_c$, although the easy-attention transformer appears to be less sensitive to the $\Delta t$ used, as previously discussed. The KNF models fail to produce accurate predictions in the cases considered. As seen in Fig. 8a, all predictions will diverge over time as small initial fluctuations in the model

accumulate over time, to analyse if the predicted velocity fields still reproduce the dynamics of the system, we compare the POD of the original test data with the predicted fields for the same data, in Fig. 8b the results are compared, showing that the predicted fields still produce similar POD modes as the original data. These results reinforce the quality of the reconstruction observed in Fig. 7 and show that the model adequately reproduces the flow dynamics, reinforcing the robustness of the methodology proposed in this work.

## Discussion

This study presents and evaluates a ROM framework based on $\beta$-VAE architectures to produce robust non-linear latent-spaces, combined with the time-prediction model obtained by means of a transformer architecture. Using a two-dimensional viscous flow around two collinear flat plates in periodic and chaotic regimes, a first analysis of the $\beta$-VAE capabilities and the latent-spaces generated using these techniques is assessed and compared with the ROM obtained through POD

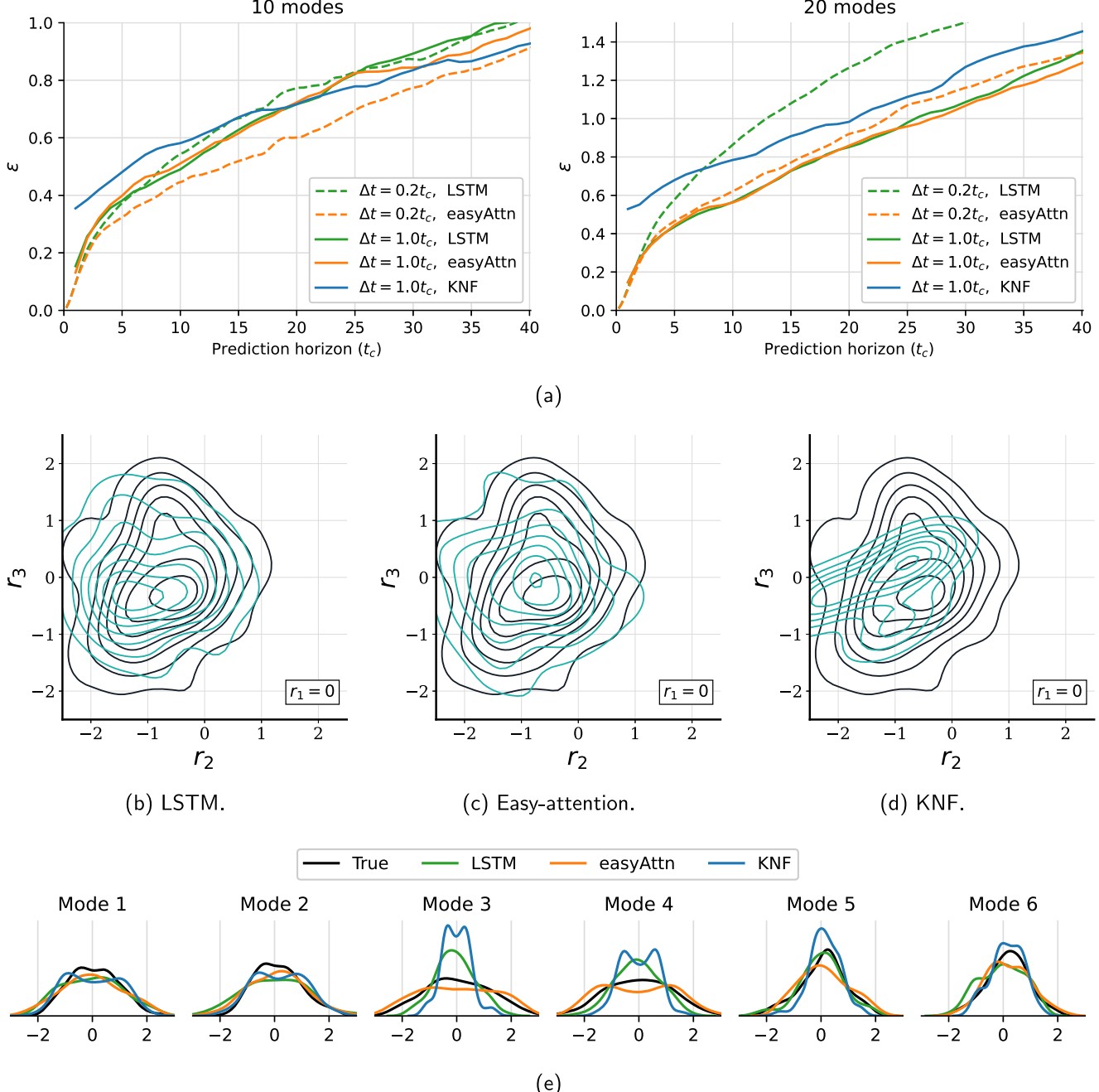

**Fig. 6 | Analysis of the predictions in latent-space.** Case with $Re = 100$, $\alpha = 80°$: (**a**) Average prediction error over the prediction horizon, **b**–**d** Poincaré sections with plane $r_1 = 0$ and $\mathrm{d}r_1/\mathrm{d}t > 0$, black lines correspond to the original data and blue lines to the predicted data. **e** Probability density functions of predicted variables. **b**–**e** Case with latent-space size $d = 20$ and $\Delta t = t_c$.

modes. For the periodic case, it is observed that the same modal features are obtained, representing a vortex shedding with an equivalent number of energetic modes. For the chaotic case, the $\beta$-VAE learns a compact near-orthogonal latent-space that significantly improves the energy reconstruction obtained by the POD using 20 modes, $E = 89.8\%$ for train data against $E = 64.4\%$ for POD. While the $\beta$-VAE ROM comprises 20 modes, a total of 69 POD modes would be required to match the result obtained by the $\beta$-VAE (in terms of flow reconstruction). The resulting modes from the $\beta$-VAE analysis exhibit temporal dynamics with shedding frequencies equivalent to those observed in the most energetic POD modes suggesting that the $\beta$-VAE is able to obtain non-linear modes that represent the most energetic or representative flow features.

The latent-space generated by the $\beta$-VAE is combined with a transformer architecture to predict the temporal dynamics. Its performance is compared against other models used in previous studies, such as LSTM or KNF. The results show that the LSTM and easy-attention transformer models are superior to the KNF model. In particular, the inherent ability of the transformer model to learn an internal representation with different frequency contents provides a more robust prediction for different time steps between snapshots compared to LSTM models. The analysis of the predictions shows that the transformer can learn the correlations among the various temporal coefficients. Combining the $\beta$-VAE and the transformer models, we obtain a ROM model that can produce predictions with $E$ of 78.1% and 64.6% at $t + t_c$ and $t + 10t_c$, respectively, for previously unseen data,

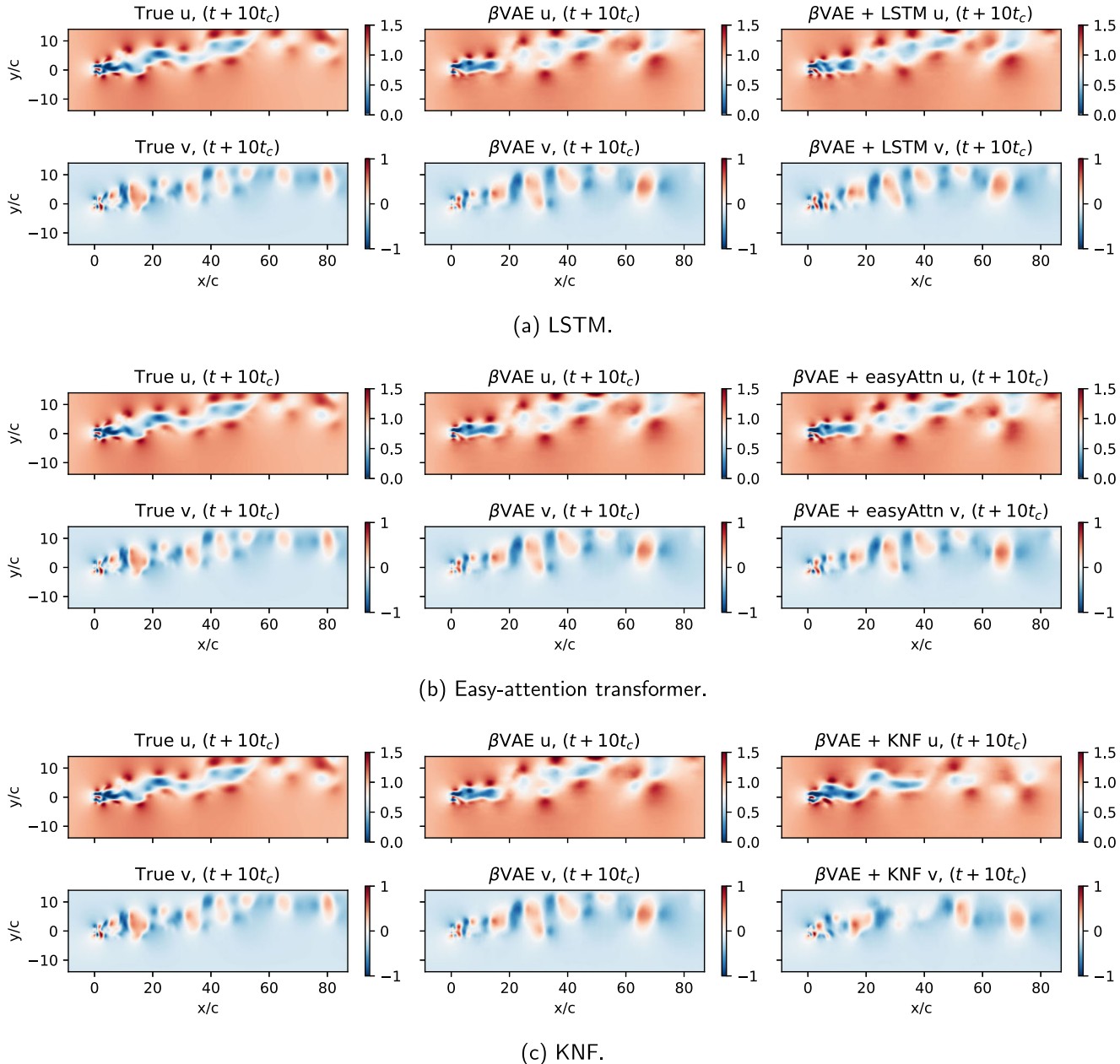

**Fig. 7 | ROM predicted fields.** True, reconstructed and predicted fields for $u$ and $v$ velocity components at $t + 10t_c$: (**a**) KNF, (**b**) LSTM, and (**c**) transformer with easy attention.

while capturing the original flow patterns. This study constitutes a proof of concept of the capabilities of these novel ML techniques to generate compact and robust non-linear ROMs that can be employed to generate predictions in chaotic flows while capturing the most relevant flow features. The combination of various VAE and transformer architectures exhibits great potential for the future of ROM development in fluid mechanics, leveraging the inherent nonlinearities of these techniques.

## Methods

### Dataset description and pre-processing

The source data set is an incompressible, two-dimensional, viscous flow over two collinear flat plates. Numerical simulation is used to solve the governing Navier–Stokes equations for this flow using an immersed-boundary projection method (IBPM)[45,46]. The two plates

have a chord length $c$ and are separated by a gap $g$ of the same size. The free stream velocity is $U_\infty$, and the Reynolds number, $Re$, is defined based on the free stream velocity and single-plate chord length. A diagram of the configuration of the flow is shown in Fig. 9c.

The simulations are performed on a series of nested grids of increasing size and decreasing resolution. The finest resolution for the simulation has a grid spacing of $\Delta x = \Delta y = 0.02c$, and the total computational domain has dimensions of $96c \times 28c$, with the upstream boundary at $x = -9c$. The dataset generation and its characteristics are described in more detail in Ref. 47. The time step in the dataset (downsampled from the simulations) was chosen to be equal to 20% of the convective time $\Delta t = c/U_\infty/5 = t_c/5$, which is sufficient to resolve the dynamics of coherent vortical structures in the wake. In order to investigate the influence of the temporal resolution in the results, a number of models were trained with one snapshot in every five, being

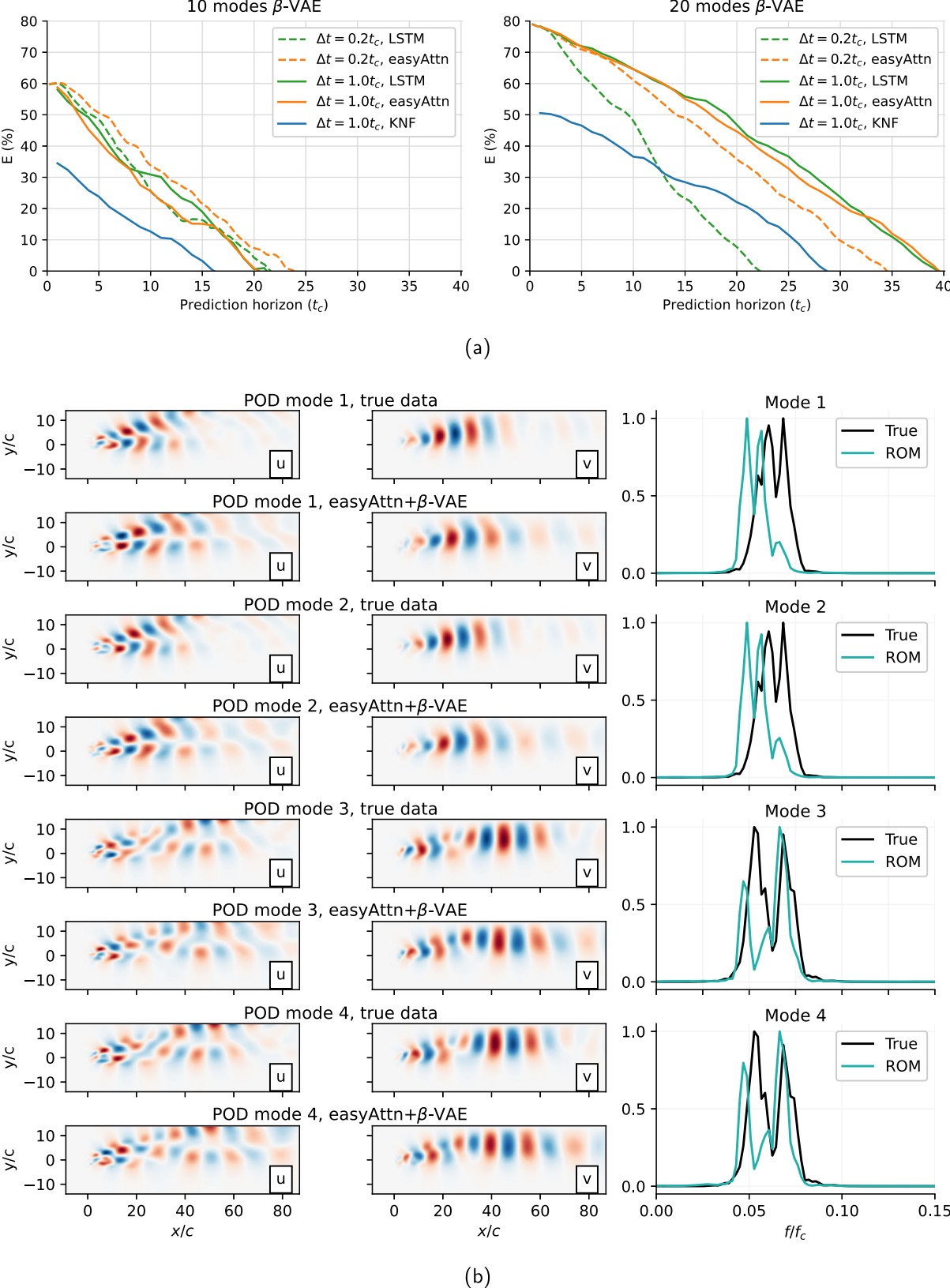

**Fig. 8 | Analysis of predicted fields. a** Average $E$ evolution over the prediction horizon. **b** POD modes comparison of true and predicted fields. This is shown for the case with $Re = 100$, $\alpha = 80°$.

$\Delta t = c/U_\infty = t_c$. For all the models, the time series of the snapshots is divided into two time series covering the respective time intervals: $[0, t_{train}]$ and $[t_{train} + 1, t_{end}]$ where $t_{train} = 27,000t_c$, which corresponds to 90% of the snapshots time series. The first period of the data is used for training, while the remaining 10% is used as test data to validate the generalisation capability of the models.

The dataset employed for this study is downsampled from the original mesh to a spatial resolution of $300 \times 98$ with a uniform grid

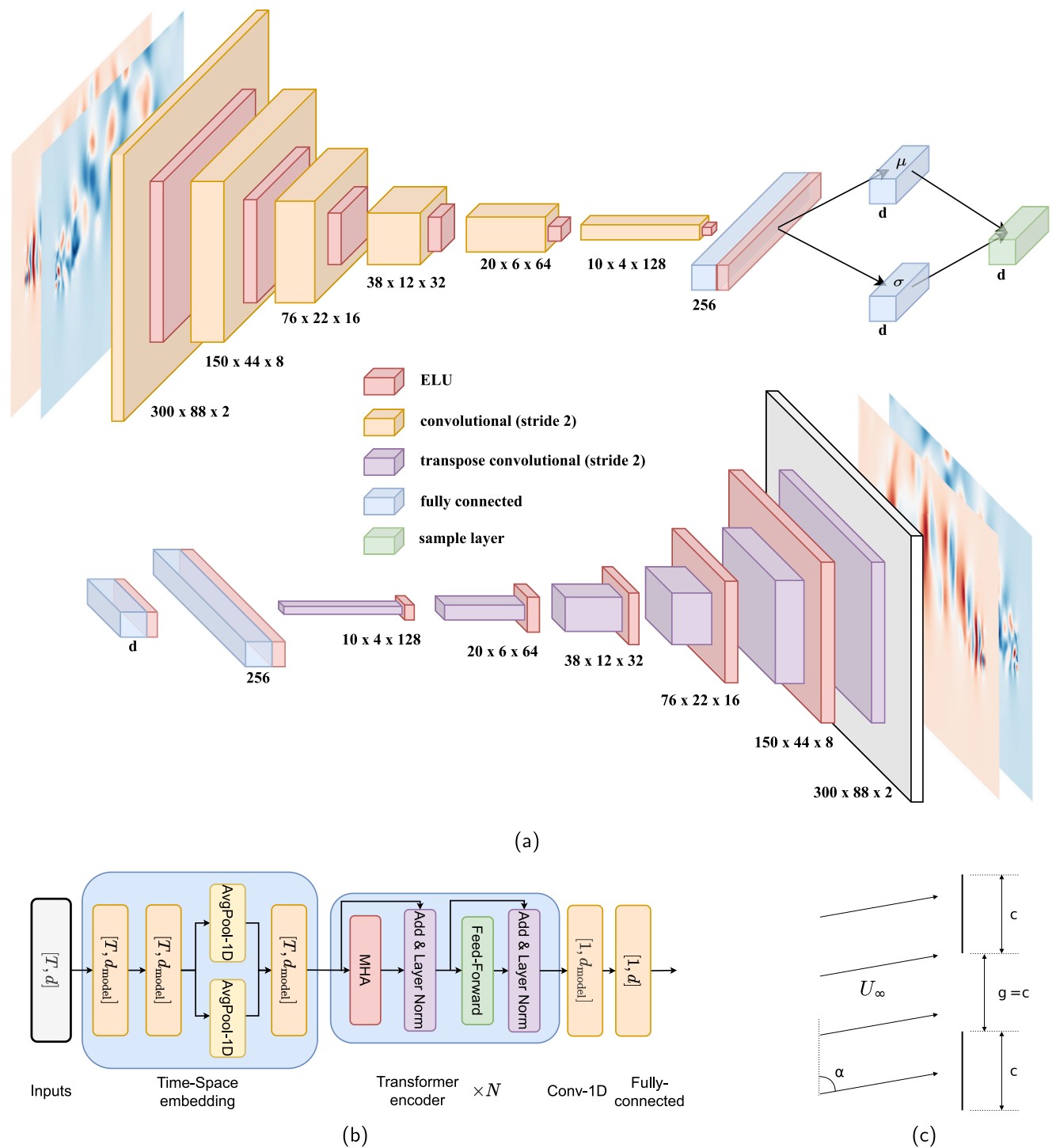

**Fig. 9 | Model architectures and case schematic. a** $\beta$-VAE encoder and decoder, (**b**) transformer. The dimension of the output for each layer has been indicated in each block. The symbols $T$ and $d_{model}$ denote the time-delay, MHA denotes multi-head attention. **c** Schematic representation of the numerical setup.

spacing, to reduce the GPU requirements for training. The fluctuation components of the streamwise $u$ and crosswise $v$ velocity were stacked as separate channels in the dataset. The values in the dataset were standardised by subtracting the pixel-wise average and dividing by the standard deviation for each velocity component over the entire dataset. The resulting dataset size was reduced to 31.7 GB and fully loaded into memory during training. This approach allowed the training process to be efficient while still capturing the essential features of the fluid flow.

**$\beta$-VAE implementation details**

The variational autoencoder (VAE) described in Kingma and Welling[48] is one of the most common architectures used in generative models. In the basic autoencoder architecture, the input **x** is directly encoded as a vector **r** using the encoder $\mathcal{E}$, $\mathbf{r} = \mathcal{E}(\mathbf{x})$, in a lower-dimensional space of size $d$, which can be mapped back to the original space by the decoder $\tilde{\mathbf{x}} = \mathcal{D}(\mathbf{r})$, being $\tilde{\mathbf{x}}$ the reconstructed output. A disadvantage of conventional autoencoders is the fact that there are no constraints imposed on the learned latent-space.

The lack of constraints in the generated latent-space can lead to overfitting and poor generalisation performance, especially for high-dimensional input data. In contrast, in VAE architectures the encoder maps input data to a parameterised prior distribution, usually a Gaussian distribution, in the latent-space $\boldsymbol{\mu}, \boldsymbol{\sigma} = \mathcal{E}(\mathbf{x})$, where $\boldsymbol{\mu}$ and $\boldsymbol{\sigma}$ denote the mean and the standard deviation, respectively. The prior Gaussian distribution encourages the model to learn a compact and smooth representation of the input data. The distribution of the latent-space is then randomly sampled, and this sample is decoded back into the original space by the decoder $\tilde{\mathbf{x}} = \mathcal{D}(\mathbf{s})$, with $\mathbf{s}$ following a distribution $\mathbf{s} \sim N(\boldsymbol{\mu}, \boldsymbol{\sigma})$. The architecture can therefore produce different outputs for the same input data, each sampled from the corresponding latent distribution. The encoder and decoder neural networks are trained by gradient descent and backpropagation, enabled by a re-parameterisation of the distribution sampling[48]. The training process for the VAE involves simultaneous training of the encoder and decoder networks with a compound loss function $\mathcal{L}$,

$$\mathcal{L}(\boldsymbol{x}) = \underbrace{\frac{1}{N_t} \sum_{i=1}^{N_t} (\boldsymbol{x} - \tilde{\boldsymbol{x}})^2}_{\text{Reconstruction loss}} - \underbrace{\frac{1}{2} \sum_{i=1}^{d} (1 + \log(\sigma_i^2) - \mu_i^2 - \sigma_i^2)}_{\text{KL loss}}, \quad (4)$$

where $N_t$ denotes the total number of points the in $\mathbf{x}$. The first term in the loss function is the reconstruction loss, $\mathcal{L}_{rec}$, which measures the model accuracy in reconstructing the input data from the reduced latent-space representation. The second term in the loss function is the Kullback–Leibler (KL) divergence loss[49], $\mathcal{L}_{KL}$, which measures the difference between the generated probability distribution and a prior probability distribution, typically Gaussian. The overall training goal is to optimise the model to produce accurate reconstructions while keeping the latent distributions close to a standard normal distribution.

The original VAE architecture was extended by Higgins et al.[50] to the $\beta$-variational autoencoder ($\beta$-VAE) architecture, which aims to promote disentangled representations in latent-space[23]. The $\beta$-VAE loss function, defined in equation (5), includes a scalar hyperparameter $\beta \geq 0$ that modulates the trade-off between reconstruction accuracy and latent-space disentanglement. A higher value of $\beta$ results in a more disentangled representation but may decrease reconstruction accuracy, whereas lower value of $\beta$ may result in a more accurate reconstruction but a less disentangled latent-space. As a result, by incorporating the $\beta$ parameter into the loss function, the $\beta$-VAE architecture can achieve a more interpretable and disentangled latent-space, which may improve the model generalisation ability.

$$\mathcal{L}(\boldsymbol{x}) = \mathcal{L}_{rec} - \frac{\beta}{2} \mathcal{L}_{KL}. \quad (5)$$

The $\beta$-VAE architecture is adapted from Ref. 25. The encoder is used to generate the temporal modes for the latent-space representation using the mean vector for each time step $\mathbf{r}(t) = \boldsymbol{\mu}(t)$. The mean vectors are generated by applying the encoder $\mathcal{E}$ to each input timestep: $\boldsymbol{\mu}(t), \boldsymbol{\sigma}(t) = \mathcal{E}(\mathbf{x}(t))$. The spatial modes are reconstructed one by one: $Y_i = \mathcal{D}(\boldsymbol{s}_i)$. Here $Y_i$ is the $i^{th}$ spatial mode, $\mathcal{D}$ is the decoder and $\boldsymbol{s}_i$ is the vector that selects the mode to be reconstructed ($\boldsymbol{s}_i = \delta_i^j = (0, \cdots, 1, \cdots, 0)$). The $\beta$-VAE architecture is sketched in Fig. 9a.

The nature of the data determines the choice of a convolutional neural network (CNN) to build the encoder, since the patterns in the flow can be better captured by convolutional layers that preserve the spatial relationships between points in the input. Fig. 9a shows a representation of the model. In the encoder, we use six convolutional layers with a stride of two so that each layer halves the spatial dimension. The spatial reduction allows the subsequent layers to capture information at larger scales in the input flow data, which also has similarities to the multiple scales of the studied chaotic flow and can help to represent it. The number of filters in each layer progressively increases to preserve the information flow while reducing the spatial dimension. After the sixth convolutional layer, the spatial information is discarded, and a fully-connected layer is added to combine the information. Finally, two parallel layers with the same number of units as the latent-space dimension are used to output the mean and variance of the latent statistical distributions. During training, the distributions are sampled to generate inputs to the decoder network, while only the $\boldsymbol{\mu}(t)$ values are used to encode the time series used later by the predictor.

The decoder model is designed as an almost symmetric network to the encoder. Latent-space samples are fed into a fully-connected layer, and its output is reshaped to the same shape as the last convolutional layer in the encoder. Six transposed convolution layers are then used to increase the spatial dimension with decreasing number of filters. A final transposed convolution layer with two filters produces the two output channels. The chosen activation function is the exponential linear unit (ELU)[51] for all layers except the last, where the activation is linear.

It is well known that chaotic flows often involve complex, non-linear interactions between fluid particles, which can lead to non-linear relationships between variables. As a result, the dominant modes of variability may not capture the complex, non-linear behaviour of the flow. This effect can be seen in Supplementary Fig. 1, where the first input to the $\beta$-VAE decoder network is sampled with different scalar values while the remaining inputs are kept to zero. The mode is sampled with input values in the range $s_1 \in [-2, 2]$ because during VAE training the latent-space is sampled from a near-standard normal distribution, driven by the KL-loss regularisation during training. The non-linear representation is evident as the shedding wake patterns change with the latent input value. This non-linear representation allows the $\beta$-VAE architecture to reproduce $E = 89.8\%$ with only 20 modes. In contrast, POD only yields $E = 64.4\%$ and would require using over 69 modes to obtain a reconstruction above $E = 90\%$.

## Proper orthogonal decomposition

In this section POD is employed as a reference to compare the resulting modes from the $\beta$-VAE latent-space with those obtained using this classical method. In particular, the snapshot method[52] has been used for the present study. Considering the streamwise and crosswise velocity components defined as $U(\mathbf{x}, t)$ and $V(\mathbf{x}, t)$, respectively, being $\mathbf{x} = (x, y)$ with $x$ and $y$ the streamwise and crosswise coordinates and $t$ the time. We decompose the velocity components as:

$$U(\mathbf{x}, t) = \overline{U}(\mathbf{x}) + u(\mathbf{x}, t), \quad V(\mathbf{x}, t) = \overline{V}(\mathbf{x}) + v(\mathbf{x}, t) \quad (6)$$

where $\overline{U}(\mathbf{x})$ and $\overline{V}(\mathbf{x})$ are the streamwise and crosswise time-averaged velocity components and $u(\mathbf{x}, t)$ and $v(\mathbf{x}, t)$ are the streamwise and crosswise fluctuating velocity components. The fluctuating quantities can be approximated as a linear combination of basis functions $\phi_i(\mathbf{x})$ as:

$$u(\mathbf{x}, t) \approx \sum_{i=1}^{N_m} a_i^u(t) \phi_i^u(\mathbf{x}), \quad v(\mathbf{x}, t) \approx \sum_{i=1}^{N_m} a_i^v(t) \phi_i^v(\mathbf{x}), \quad (7)$$

where $a_i(t)$ are time-dependent coefficients and $N_m$ is the number of basis functions used. Here we assume a number of images $N_t$, each one consisting of $N_p$ grid points along the spatial domain $\mathbf{x}$, with $N_t < N_p$. Following the snapshot method[52], each image can be treated as an $N_p$-

dimensional vector and the data can be arranged into an $N_t \times N_p$ snapshot matrix:

$$\mathbf{u} = \begin{bmatrix} u(x_1,t_1) & \cdots & u(x_{N_p},t_1) \\ \vdots & \ddots & \vdots \\ u(x_1,t_{N_t}) & \cdots & u(x_{N_p},t_{N_t}) \end{bmatrix} ; \mathbf{v} = \begin{bmatrix} v(x_1,t_1) & \cdots & v(x_{N_p},t_1) \\ \vdots & \ddots & \vdots \\ v(x_1,t_{N_t}) & \cdots & v(x_{N_p},t_{N_t}) \end{bmatrix}.$$

(8)

Using the snapshot matrix, the two-point correlation matrix can be written as $\mathbf{G} = \mathbf{u}\mathbf{u}^T + \mathbf{v}\mathbf{v}^T$, where the superscript $^T$ refers to the matrix transpose. Solving the eigenvalue problem of $\mathbf{G}$ returns the eigenvalues $\lambda_i$ and the left and right eigenvector matrices. The left and right eigenvector matrices are respectively the matrix $\mathbf{\Psi}$ containing in its columns the temporal modes $a_i(t)$ (which are orthonormal vectors of length $N_t$ and unitary norm) and its inverse (i.e. its transpose). Note that the columns of $\mathbf{\Psi}$ form a basis of rank $N_t$ and that the eigenvalues $\lambda_i$ are representative of the energy contribution of each mode. The orthonormal spatial modes $\phi_n(\mathbf{x})$ can then easily be computed as $\mathbf{\Sigma}_u \boldsymbol{\phi_u} = \mathbf{\Psi}^T \mathbf{u}$ and $\mathbf{\Sigma}_v \boldsymbol{\phi_v} = \mathbf{\Psi}^T \mathbf{v}$, where $\mathbf{\Sigma}_u$ and $\mathbf{\Sigma}_v$ are diagonal matrices which, in each $i^{th}$ diagonal element, contain the streamwise and wall-normal Reynolds-stress contributions of the $i^{th}$ mode.

For consistency purposes, the POD is computed using the snapshots time series that covers the time interval $[0, t_{train}]$.

### Time-series prediction models
The $\beta$-VAE encoder network generates the entire dataset latent-space time series, $\mathbf{r}(t)$, where the snapshots time series is divided into two time series that cover the following time intervals: $[0, t_{train}]$ and $[t_{train} + 1, t_{end}]$ where $t_{train} = 27,000 t_c$ that corresponds to the 90% of the snapshots time series. The first period of the data is used for training being the last 10% used as test data. Note that the test data have not been used for training the $\beta$-VAE and is only encoded by the previously trained $\beta$-VAE model. The transformer model is used to predict the latent-space vector $\mathbf{r}(t+1)$. In the present study, we use the time-delay[53] dimension of 64 steps for temporal prediction, which means that the input to the transformer is a sequence of the previous 64 time steps, and the output is the prediction of the next time step for the latent vector in the sequence. The transformer is trained to minimise the difference between the prediction of the next time step and the true data using a mean-squared-error loss function.

A time-space embedding module is added to each input latent-space vector to incorporate temporal and spatial information before passing it to the transformer blocks, allowing the model to distinguish between latent vectors generated at different time steps. The pooling layers are designed to draw the characteristic information from time-series data, facilitating the model to capture the key information of temporal dynamics of the physical system. Note that we adopt stride steps of two for one-dimensional average pooling and maximum pooling.

The transformer model comprises a stack of transformer encoder blocks, each consisting of a multi-head attention block and a feed-forward neural network. Note that, in the present study, we employ two types of attention mechanisms: self-attention[28] and easy attention[43], which has demonstrated promising performance in predicting the temporal dynamics of chaotic systems, and in our case significantly outperforms self-attention transformer. The attention blocks allow the model to weigh the importance of different parts of the input sequence when making predictions[54], while the feed-forward network allows it to learn complex non-linear relationships between the input and output sequences. In the present study, we use four heads for attention modules to implement multi-head attention and adopt a feed-forward dimension of 128. We adopt four transformer blocks to ensure the capability to identify the complex dynamics in latent-space. After the transformer blocks, a one-dimensional convolutional network and a fully-connected layer are added to decode the transformer output and form the final latent-space vector prediction. The architecture is illustrated in Fig. 9b.

The LSTM model architecture includes four layers of LSTM elements, followed by a fully-connected layer of 128 neurons and a final output layer matching the latent-space size. The time-delay dimension for the LSTM is the same as transformer models.

The KNF-model implementation is based in the code from Ref. 27. After the hyperparameter tuning, the number of previous time steps used to predict is 5, and the maximum order of the functions for construction of the forcing term is 3 for polynomial functions and 4 for trigonometric functions.

### Training setup
The Torch 2.0 deep-learning framework[55] was used to implement the models and the training pipeline. An NVIDIA GeForce RTX 4090 and an NVIDIA A100 GPU were used to train the $\beta$-VAE models and transformer models, respectively. Training the $\beta$-VAE model and transformer took approximately 40 minutes and 100 minutes, respectively. The $\beta$-VAE model is trained using the Adam algorithm[56]. The learning rate is variable with a one-cycle schedule as proposed in[57], starting at $1 \times 10^{-4}$, with a maximum value of $2 \times 10^{-4}$ at 20% of the training epochs and decreasing to $5 \times 10^{-6}$ at the end of the training. The model was trained over 1000 epochs using batch size of 256. The encoder and decoder network have $1.06 \times 10^6$ trainable parameters each. The first 90% of the snapshots time series are used for training, and the remaining are used for testing the models.

The resulting temporal dynamics from the $\beta$-VAE are then used to train a transformer architecture using the Adam algorithm[56] with $\epsilon$ of $1 \times 10^{-8}$ for stability reasons. The learning rate was initially set to $1 \times 10^{-3}$ and decreased to $6.6 \times 10^{-6}$ within 1,000 epochs via exponential decay using a decay rate of 0.99, whereas the batch size was set to 256. Note that for the case with a sampling factor of 5, we set the early-stopping schedule with respect to the loss, which stops the training process after 50 epochs if the error value is no longer decreasing. Table 1 summarises the employed architectures in the present study. The last 10% of time steps are not used during training are utilised as test data.

### Data availability
All datasets used in this study are openly available in Zenodo, accessible at: https://doi.org/10.5281/zenodo.10501215.

### Code availability
The codes used for this work are available at: https://github.com/KTH-FlowAI/beta-Variational-autoencoders-and-transformers-for-reduced-order-modelling-of-fluid-flows.

**Table 1 | Summary of the architectures employed in the time-series prediction**

| Name | Self | Easy | LSTM |
|---|---|---|---|
| Time-delay (T) | 64 | 64 | 64 |
| $d_{model}$ | 64 | 64 | – |
| FFD/Hidden | 128 | 128 | 128 |
| No.heads | 4 | 4 | – |
| Num layers | 4 | 4 | 4 |
| No.Parameters | $1.45 \times 10^5$ | $1.59 \times 10^5$ | $4.79 \times 10^5$ |

We denote the size of time delay as *T* and the embedding size as $d_{model}$, respectively. Note that Feed-Forward denotes the dimension of the feed-forward network in the transformer encoder while the hidden-state dimension of the LTSM layer is denoted as Hidden-State, respectively.

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

## Acknowledgements
The authors would like to thank to Dr. Hamidreza Eivazi and Mr. Samuel Molina for their technical assistance in the first part of the study and Dr. Steve Brunton for his insightful suggestions. Some of the deep-learning models were trained by means of resources provided by the National Academic Infrastructure for Supercomputing in Sweden (NAISS). RV acknowledges the financial support from ERC grant no. '2021-CoG-101043998, DEEPCONTROL'. Views and opinions expressed are however those of the author(s) only and do not necessarily reflect those of the European Union or the European Research Council. Neither the European Union nor the granting authority can be held responsible for them.

## Author contributions
ASR: Methodology, Software, Validation, Investigation, Data Curation, Writing – Original Draft, Writing – Review & Editing, Visualisation. CSV: Methodology, Software, Validation, Investigation, Data Curation, Writing – Original Draft, Writing – Review & Editing, Visualisation. MAG: Methodology, Software, Writing – Review & Editing. YW: Methodology, Software, Validation, Writing – Review & Editing. AA: Data generation. SD: Data generation, Writing – Original Draft, Writing – Review & Editing RV: Ideation, Methodology, Writing – Review & Editing, Supervision, Resources, Funding acquisition.

## Funding

## Competing interests
The authors declare no competing interests.
