## [Peer Review File · Nature Communications]

β -Variational autoencoders and transformers for reduced-order modelling of fluid flowsReviewers' Comments:

Reviewer #1:

Remarks to the Author:

This research on the coupling of VAEs to Transformers for fluid mechanics deserves investigation. But the results proposed in this paper suffer from weaknesses.

1- A plot of the fraction of energy reconstructed by the VAE's latent-variables is missing for the training data.

2- The Poincaré map is, in principle, a good idea, but why did the authors decide to plot this map for r_6 and r_{10} ? The most important latent variables are r_1 and r_2 . The reader should understand why there is no plot on the most important latent variables.

3- Why the prediction of the velocity fields for $t+t_c$ and $t+5t_c$ in Figure 10 is not performed using VAE+LSTM nor VAE+KNF?

4- The results obtained with $\beta=0$ (classical AE) are missing for a discussion on the "near-orthogonality" of the latent variables.

5- VAE should not be trained using the full data set. If you use the full data, you cannot discuss the generalization of neural networks.

Detailed comments

Page 1: Add more references on VAE for fluid flow predictions. For example : A Bayesian Nonlinear Reduced Order Modeling Using Variational AutoEncoders, Nissrine Akkari , Fabien Casenave , Elie Hachem , David Ryckelynck, Fluids, 2022, 7 (10), pp.334. ([10.3390/fluids7100334](https://doi.org/10.3390/fluids7100334)).

Page 1: Give a comment on the role of the beta parameter of VAE. The one given on page 4, should be placed on page 1: "The value of β is chosen high enough to produce a near-orthogonal latent-space representation but as small as possible to avoid increasing the reconstruction error."

Page 1: It is not at all clear what you are calling a "mode" for VAE. The definition comes only at the end of page 3.

Page 2: There are no results in the Results section!

Page 2 : Equation (1), why the index i ?

Page 3 : Equation (3) what is the index k in E_k ?

Page 3 : What are r_i and a_i ?

Page 5: Figure 3, there is almost no improvement in reconstruction loss, only a reduction factor around 2 after training. What does "Train step" mean?

Page 10: The title of figure 9 is not consistent. Change r_0 to r_1 .

Reviewer #2:

Remarks to the Author:

This work proposes a new ML-based ROM for fluid flows using a β -VAE for spatial low-dimensionalization and a Transformer for temporal prediction of the latent dynamics.

The result using the proposed method for spatial low-dimensionalization is compared with the conventional POD, and it is clearly demonstrated that the proposed method can reconstruct the original chaotic flow field much less number of modes than the POD.

As for the temporal prediction, too, a much better performance than LSTM and KNF is clearly demonstrated.

The manuscript is excellently written (except for some minor points detailed below), and I could go through the manuscript without feeling stress.

The sensitivity study on the β parameter is well documented.

I think that the proposed method can be applied not only to more complex fluid flow problems but also to diverse nonlinear dynamics problems, and the present work will attract substantial attentions of the readers of Nat. Commun.

Therefore, I would like to recommend this work for publication in Nat. Commun. after the authors consider some minor points below and make a revision reflecting them.

1. Eqs. (1) and (3): Instead of $\times 100$, $\times 100\%$ (or removing $\times 100$) will make more sense as 100% is unity. Similarly, "multiplied by 100" in the 3 lines below Eq. (1) looks weird and inconsistent with "A value of 1" below Eq. (2).

2. Flow configurations described on Page 3: It is hard to understand the flow configurations without figures, although I can roughly guess from Figs. 2, 4 etc.

3. Page 3: I am a bit confused by the meaning of the subscript i to \mathbf{s} , \mathbf{r} , and \mathbf{a} , because it is added to the bold-face vectors. Doesn't i represent the mode number? I cannot understand the meaning of "the desired i^{th} position" on the same page --- what does it mean by "position"? The same applies to \mathbf{r}_i on Page 5.

4. Related to the comment 3 above, I am curious to see how the trajectory of the latent variables is for the periodic case.

5. Fig. 8(a): The instantaneous behaviors are not well reproduced by any models, which is somehow typical to nonlinear cases. But I am curious whether the present Transformer model works well for the periodic case. For instance, Hasegawa et al. (Fluid Dyn. Res. 2020) reports the LSTM-based ROM suffers from a substantial L2 error even in a periodic case due to a small phase error accumulating every step.

6. Related to the comment 5 above, "The transformer produces accurate predictions" on Page 9 may better be expressed as "The transformer produces more accurate predictions".

7. Pages 12-13: From my own experience, the network structure and its depth used in the present study looks reasonable for the present flow problems. But can the authors note any comments when the flow becomes more complex, e.g. turbulent channel flows?

8. Page 13: I am just curious why the authors chose ELU, although there are some previous studies (e.g., Fukami et al., SN Comput. Sci. 2021) which show that ELU (when used in CNN-AE) is relatively robust against different flow problems.

9. Page 13, 2nd line from the bottom: I cannot understand "The last 1024 time steps are then used as test data" which, for the periodic case, looks to exceed "1,000 instantaneous flow fields" on Page 3.

Reviewer #3:

Remarks to the Author:

The authors present a β -VAE framework to identify the modes of two relatively simple canonical test cases of flow dynamics. The authors then predict the evolution of the flow with a transformer architecture. I do not think that the paper in its current form can be published in this journal.

Major general concerns:

1. Novelty: This is an interesting paper, but nothing substantially novel to demand a publication in Nat Comm. The authors used a well-established VAE architecture to identify a latent space that is argued to represent the true modes of the system. It is validated by a very simple periodic example that is not a turbulent flow. Then the authors use another well-established transformer architecture to predict the evolution of the modes. Again, there is no novelty there. Besides, the time series forecasts shown by the authors do not reflect how transformers are doing substantially better than the other baselines, which show pretty low-skill in themselves. Fig 8b shows transformer has a larger error than the other two baselines. Also, why does KNF start off with a bump in that figure?

2. Assumptions: There has been a lot of previous work in POD-based decomposition and then prediction; AE-based decomposition and then prediction, etc, etc. Studies looking at these canonical systems often ignore the fact the real turbulent flow at $Re > 10K$ or so has much more complexity. And more importantly, the relevant modes in those systems, arising due to the non-normality of the flow, are not orthogonal-- a central assumption in this paper, following many other studies. This has huge implications and has been studied in other fields of turbulence like geophysical flows; see 1. <https://rmets.onlinelibrary.wiley.com/doi/abs/10.1002/qj.305> 2. <https://journals.ametsoc.org/view/journals/atsc/73/9/jas-d-16-0099.1.xml> .

3. State-of-the-art (SOTA): Data-driven prediction of turbulent flow fields has seen a lot of progress in the geophysical flows community. For example, data-driven architectures can predict real weather dynamics (not canonical systems) at a much higher resolution (than what is studied in this paper) and outperform numerical simulations. These are real systems with SOTA archs achieving better performance. See 1. <https://arxiv.org/abs/2212.12794>; 2. <https://arxiv.org/abs/2211.02556>; 3. <https://arxiv.org/abs/2202.11214>. Even in canonical turbulent flow with high Re (both 5k and 10K), other authors have shown SOTA results, see 1. <https://arxiv.org/abs/2010.08895> 2. <https://openreview.net/forum?id=1C36tFZn7sR>

Looking at these results, it seems that one can actually predict turbulent flow (at a much higher Re compared to this study) without resorting to "just" predicting the temporal evolution of "modes". This raises the question as to why we must even go to this route except for the fact that there are many other studies that do this using some flavor of AE to extract some modes and some flavor of a temporal integrator to predict the modes for simple systems.

I still believe there is a lot of value in fundamental studies looking at modal decompositions with incremental skill but improved interpretability, yet this particular paper does not have the novelty or address any fundamental theory to be published in Nat Comm. I feel it is more suitable elsewhere in a more technical journal.

Major Technical Comments:

1. Figure 5 shows 10 modes. What is the true mode corresponding to these predictions? What is the reason to believe that they are close to the true mode? In this simple example, I understand that the authors have a general sense of how a mode might look like, since it differs ever-so-slightly from the periodic case, but how does the VAE framework help us study a more general setting with high Re ? Is this framework generalizable in that sense?

2. Fig 8 (a) and (b). From (a) I feel that all the methods are doing as well or as poorly. Then why is there an emphasis on the transformer? From (b), it feels that the transformer is doing worse(?) Also, why does the KNF have a big jump at the first time step in Fig 8(b)?

3. Fig 9 is actually really good. It shows the advantage of having a transformer. But still, the Poincare section does not correspond to the truth although it spans a larger space of the truth. If I were to guess, all three methods are suffering from the fact the β -VAE modes are not really representing the most relevant dynamical modes. Also, what is the rationale behind them being the most physical

modes? The fact that the authors made them orthogonal? But that's just as bad as POD modes-- isn't it?

4. Does the long predicted trajectory have the same POD modes as the truth? It's a simple test to study first-order variability? How about the PDF?

5. There is a lot of emphasis on visualization in the paper, but there is not enough conviction that the modes visualized are the right modes if we don't know the true modes of the system (which is obviously close to impossible for a real system, but even for this one?)

6. Finally, I think the system chosen is too canonical and has little similarity to large-scale practical systems and hence many of the arguments might be too fine-tuned to this toy problem.

Comments to Reviewer #1:

The authors would like to thank the Reviewer for their comments and thorough examination of our work. The Reviewer's comments are reproduced below, with our responses following each comment in blue text. The changes are highlighted in blue in the text. The paper has been revised, taking the outlined changes into account; thus, we are attaching a non-marked-up version to allow easier reading of the manuscript. Please note that the numbering follows the marked-up version when citing references and figures in this rebuttal.

Before proceeding with the rebuttal, we would like to remark that we have substantially reworked the discussion of the results following the referees' comments.

Reviewer #1 (Remarks to the Author):

This research on the coupling of VAEs to Transformers for fluid mechanics deserves investigation. But the results proposed in this paper suffer from weaknesses.

We thank the Reviewer for their interest in the topic of the present work. We have added results and discussion to improve the paper's quality and hope that we can address all the comments below to their satisfaction.

In this study, some analyses and procedures have not been reported in the previous literature, and some insights can be helpful for the community to improve future applications of autoencoder approaches to perform reduced-order modelling. We have improved the manuscript's quality and hope to address all the reviewer's concerns. In particular, we would like to highlight the following inputs:

- We have extended our numerical simulation data to 150,000 snapshots with an interval between snapshots of 0.2 convective times. This database has allowed us to perform a detailed analysis comparing the influence of the number of snapshots and dynamic cycles to generalise the latent space. In addition, this database has also allowed us to compare the performance of the different neural-network architectures in the time-series forecasting for different intervals between snapshots and different time-scale ranges.
- In most prior studies based on data-driven approaches, the splitting data procedure is performed using random splitting, and therefore, the obtained ROMs and metrics are biased by the fact that the model has observed information of the complete flow dynamics (see the following examples: Murata, T., Fukami, K., & Fukagata, K. (2020). *Nonlinear mode decomposition with convolutional neural networks for fluid dynamics*. Journal of Fluid Mechanics, 882, A13; Akkari, N., Casenave, F., Hachem, E., & Ryckelynck, D. (2022).; Maulik, R., Lusch, B., & Balaprakash, P. (2021). *Reduced-order modelling of advection-dominated systems with recurrent neural networks and convolutional autoencoders*. Physics of Fluids, 33(3)). In our study, we took an approach based on splitting the temporal dynamics into two intervals: $[0, t]$ and $[t+1:t_{end}]$, so the β -VAE is trained using only information from the first part of the numerical simulation. We prove that the latent spaces we obtained are robust and are representative of the physics even in snapshots unrelated to the training split due

to the chaotic nature of our example. The generalisation was achieved by choosing an appropriate model complexity that better balances the bias–variance tradeoff and by traditional hyperparameter tuning.

- Our study presents an entirely data-driven approach, i.e. that does not rely on auxiliary partial differential equations. Furthermore, our method has been tested in a complex flow, namely the 2D incompressible, two-dimensional, viscous flow around two collinear flat plates. This configuration leads to chaotic flow described by the Navier-Stokes equations, and constitutes a significant step forward with respect to the classic 2D low-Reynolds-number cylinder flow (which is not chaotic) or the Burgers equation, which have been used in most of the studies in the literature. Therefore, we argue that the flow we analyze is significantly complex, beyond previous works relying on simplified “toy problems”.
- In this study, modern transformer architectures have been tested, such as: the novel Easy Attention (see *Sanchis-Agudo, M., Wang, Y., Duraisamy, K., & Vinuesa, R. (2023). Easy attention: A simple self-attention mechanism for Transformers. arXiv preprint arXiv:2308.12874*). This method has been proved to be an improvement in comparison with the standard Self-Attention approach (which we used in the first version of the manuscript). Instead of relying on keys, queries and values to compute the attention score, the easy-attention mechanism directly treats the attention scores as learnable parameters, a fact that enables the model to capture the long-term temporal dynamics in chaotic systems.

Figure: Comparison of average prediction error over the prediction horizon in latent space, for self-Attention and easy-attention transformers.

- Moreover, we have tested several ROMs developed using different time steps. We have observed that the Transformer-based architectures are more efficient than the classical LSTM when the range of frequencies involved in the prediction is high. This can be explained by the ability of the transformers to learn phenomena associated with different time scales. These analyses are performed not in terms of reconstruction error, but also using Poincaré maps. This allows us to analyse the robustness of the state prediction, in order to show the potential of these approaches to work even as a potential state-estimator model, as reported with other approaches based on Kalman filters (see *Colburn, C. H., Cessna, J. B., & Bewley, T. R. (2011). State estimation in wall-bounded flow systems. Part 3. The ensemble Kalman filter.*

Journal of Fluid Mechanics, 682, 289-303). These models are helpful to tackle the problem of controlling a turbulent flow based on its estimated state.

1- A plot of the fraction of energy reconstructed by the VAE's latent-variables is missing for the training data.

We thank the Reviewer for their suggestion, and we have included the corresponding figures with training and testing data. Our database has been extended to 150,000 snapshots with a time step of 0.2 convective time units in order to ensure a convergence of the latent space in both train and test datasets. Now the temporal dynamics, and all the analyses reported in this study, come from data predicted from our model that has not been used in any phase of the training process (as opposed to what is typically done in the literature).

Additionally, an analysis of the β effect and the latent-space dimension over the test/train metrics of the latent space has been performed and the results are described in the text.

Figure. Fraction of E reconstructed by POD and β -VAE as a function of the number of modes. Left: $Re = 40$, $\alpha = 90^\circ$. Right: $Re = 100$, $\alpha = 80^\circ$.

2- The Poincaré map is, in principle, a good idea, but why did the authors decide to plot this map for r_6 and r_{10} ? The most important latent variables are r_1 and r_2 . The reader should understand why there is no plot on the most important latent variables.

We thank the referee for this point. Indeed, these are the most relevant latent variables, and therefore, we have performed the section using the most energetic mode r_1 and plotted the following pair of most energetic modes r_2 and r_3 . We just chose an example of a combination. We have included this plot in the manuscript, and we show all the combinations in the extended material to show the reader that all the Poincaré maps are adequate for all the latent variables.

Figure. Poincaré sections for r_2 and r_3 with plane $r_1 = 0$ and $dr_1/dt > 0$. Black lines correspond to the original data and blue lines to the predicted data. From left to right: KNF, LSTM, Transformer.

3- Why the prediction of the velocity fields for $t+t_c$ and $t+5t_c$ in Figure 10 is not performed using VAE+LSTM nor VAE+KNF?

We thank the Reviewer for this suggestion, and we have added the VAE+KNF in the comparison. We also increased the prediction length to $10t_c$.

4- The results obtained with $\beta=0$ (classical AE) are missing for a discussion on the "near-orthogonality" of the latent variables.

We have trained a classical VAE with $\beta=1$ and another one with $\beta=0$. Additionally, we have adapted the architecture to classical AE, which is not only $\beta=0$ since it is necessary to replace the distribution sampling in the bottleneck for a simple dense layer that encodes the data. As a result, we have shown that, as expected by previous results (see Burgess, C. P., Higgins, I., Pal, A., Matthey, L., Watters, N., Desjardins, G., & Lerchner, A. (2018). Understanding disentangling in β -VAE. arXiv preprint arXiv:1804.03599), the Variational Autoencoders are able to generalise better with respect to the classical AE. Moreover, the β -VAE, with the ability to fine tune the β parameter, allows us to find robust latent spaces in both: train and test data.

Method	MSE Train	MSE Test
β -VAE 20-modes ($\beta=0.05$)	0.0716	0.177
β -VAE 20-modes ($\beta=0$)	0.0387	0.2159
VAE 20-modes ($\beta=1$)	0.2612	0.3305
AE 20-modes	0.0408	0.2345

With these analyses, we highlight the relevance of the β -VAE architecture to get latent spaces preventing overfitting issues that can be used as robust data representations in lower-dimensional spaces being the more relevant result of the ability of generalisation of the latent spaces.

5- VAE should not be trained using the full data set. If you use the full data, you cannot discuss the generalisation of neural networks.

We thank the reviewer for raising this point, as it allows us to clarify an important concept: generalisation of the latent spaces for both training and testing data. It is worth noting that now our VAE is trained on the first 90% of the dataset, leaving the remaining 10% for testing. This split method ensures that the training data does not include snapshots close to the testing data samples.

We have extended the dataset to perform these analyses, finding that learning a robust latent space requires a large number of dynamic cycles, and not only a large number of snapshots. This can be observed in the following table, where the VAE error metrics

obtained for both cases are similar, despite the fact that the VAE model trained with $1t_c$ was trained with only one snapshot in every five.

Method	E Train $1t_c$	E Test $1t_c$	E Train $0.2t_c$	E Test $0.2t_c$
β -VAE 20-modes	90%	80%	91%	76%
β -VAE 10-modes	82%	62%	86%	58%
POD 10-modes	45%		45%	
POD 20-modes	63%		63%	

Detailed comments

Page 1: Add more references on VAE for fluid flow predictions. For example : A Bayesian Nonlinear

Reduced Order Modeling Using Variational AutoEncoders, Nissrine Akkari , Fabien Casenave , Elie Hachem , David Ryckelynck, Fluids, 2022, 7 (10), pp.334. <10.3390/fluides7100334>.

We thank the Reviewer for their suggestion, and we have added this reference and a few more recent studies related to the topic.

- Akkari, N., Casenave, F., Hachem, E., & Ryckelynck, D. (2022). A bayesian nonlinear reduced order modeling using variational autoencoders. Fluids, 7(10), 334.
- Maulik, R., Lusch, B., & Balaprakash, P. (2021). Reduced-order modeling of advection-dominated systems with recurrent neural networks and convolutional autoencoders. Physics of Fluids, 33(3).
- Luo, Z., Wang, L., Xu, J., Chen, M., Yuan, J., & Tan, A. C. (2023). Flow reconstruction from sparse sensors based on reduced-order autoencoder state estimation. Physics of Fluids, 35(7).
- Zhang, B. (2023). Nonlinear mode decomposition via physics-assimilated convolutional autoencoder for unsteady flows over an airfoil. Physics of Fluids, 35(9).
- Raj, N. A., Tafti, D., & Muralidhar, N. (2023). Comparison of reduced order models based on dynamic mode decomposition and deep learning for predicting chaotic flow in a random arrangement of cylinders. Physics of Fluids, 35(7).

Due to the word limitation (5,000 words), we could not include a detailed description of every study, but we hope that the inclusion of the previous references would help the reader to have a view of the state-of-the-art.

Page 1: Give a comment on the role of the beta parameter of VAE. The one given on page 4, should be placed on page 1: "The value of β is chosen high enough to produce a

near-orthogonal latent-space representation but as small as possible to avoid increasing the reconstruction error.”

We thank the Reviewer for their suggestion and we have extended the discussion of the influence of β according to our previous and new results.

Page 1: It is not at all clear what you are calling a “mode” for VAE. The definition comes only at the end of page 3.

We thank the Reviewer for their suggestion. We defined these concepts in the Methods section, which can confuse the reader. For this reason, we clarified previous references and added this clarification:

The β -VAE spatial modes are defined as the result of using the β -VAE decoder network with an input vector, s_i , containing a unit value in the desired i th position and zero elsewhere, $s_i = \delta_{ij} = (0, \dots, 1, \dots, 0)$, that gives as a result the corresponding spatial i^{th} mode. For a detailed description, the reader is referred to the Methods section.

Page 2: There are no results in the Results section!

We apologise for the confusion. We have reorganised the text, incorporating the metrics definition to the following subsection.

Page 2 : Equation (1), why the index i ?

We thank the Reviewer for their observation. We have removed the previous index i , as it is not needed to define the metric.

Page 3 : Equation (3) what is the index k in E_k ?

We have renamed the variable E_k as E to avoid misunderstandings.

Page 3 : What are r_i and a_i ?

We thank the Reviewer for their observation. We have revised and renamed some variables to make the manuscript clearer:

r_i , renamed to $r_i(t)$, refers to the i^{th} time series of latent space variables.

$a_i(t)$ refers to the i^{th} time-dependent POD coefficient series, as defined in the Methods section.

s refers to a vector in the latent-space to be decoded, to analyse the latent space, for example.

s_i refers to a one-hot encoded vector in the latent-space to be decoded, that selects the mode to be individually decoded.

Page 5: Figure 3, there is almost no improvement in reconstruction loss, only a reduction factor around 2 after training. What does "Train step" mean?

We apologise for the confusion. The reconstruction loss starts with a value of 1 but rapidly decreases in the first 20 epochs. The y-axis was limited to a maximum value of 0.5. We have adapted the Figure to make sure everything is clear.

Figure : Evolution of β -VAE losses during training: (left) 10 modes, (right) 20 modes, (solid) train (dashed) test. Losses are defined in the Methods section.

Page 10: The title of figure 9 is not consistent. Change r0 to r1.

We thank the reviewer for raising this issue, we have adapted the Figure and the index associated with it.

Comments to Reviewer #2:

The authors would like to thank the Reviewer for their comments and thorough examination of our work. The Reviewer's comments are reproduced below, with our responses following each comment in blue text. The changes are highlighted in blue in the text. The paper has been revised, taking the outlined changes into account; thus, we are attaching a non-marked-up version to allow easier reading of the manuscript. Please note that the numbering follows the marked-up version when citing references and figures in this rebuttal.

Before proceeding with the rebuttal, we would like to remark that we have substantially reworked the discussion of the results following the referees' comments.

Reviewer #2 (Remarks to the Author):

The authors would like to thank the Reviewer for their comments and thorough examination of our work. The Reviewer's comments are reproduced below, with our responses following each comment in blue text. The changes are highlighted in blue in the text. The paper has been revised, taking the outlined changes into account; thus, we are attaching a non-marked-up version to allow easier reading of the manuscript. Please note that the numbering follows the marked-up version when citing references and figures in this rebuttal.

Before proceeding with the rebuttal, we would like to remark that we have substantially reworked the discussion of the results following the referees' comments.

This work proposes a new ML-based ROM for fluid flows using a β -VAE for spatial low-dimensionalization and a Transformer for temporal prediction of the latent dynamics. The result using the proposed method for spatial low-dimensionalization is compared with the conventional POD, and it is clearly demonstrated that the proposed method can reconstruct the original chaotic flow field much less number of modes than the POD. As for the temporal prediction, too, a much better performance than LSTM and KNF is clearly demonstrated. The manuscript is excellently written (except for some minor points detailed below), and I could go through the manuscript without feeling stress. The sensitivity study on the β parameter is well documented. I think that the proposed method can be applied not only to more complex fluid flow problems but also to diverse nonlinear dynamics problems, and the present work will attract substantial attentions of the readers of Nat. Commun. Therefore, I would like to recommend this work for publication in Nat. Commun. after the authors consider some minor points below and make a revision reflecting them.

We thank the Reviewer for their favourable judgement on the interest of the presented work, and we hope that we could address all the comments below to their satisfaction.

We have been improving the manuscript's quality and hope to address all the reviewer's concerns. In particular, we would like to highlight the following inputs:

- We have extended our numerical simulation data to 150,000 snapshots with an interval between snapshots of 0.2 convective times. This database has allowed us to perform a detailed analysis comparing the influence of the number of snapshots and dynamic cycles to generalise the latent space. In addition, this database has also

allowed us to compare the performance of the different neural-network architectures in the time-series forecasting for different intervals between snapshots and different time-scale ranges.

- In most prior studies based on data-driven approaches, the splitting data procedure is performed using random splitting, and therefore, the obtained ROMs and metrics are biased by the fact that the model has observed information of the complete flow dynamics (see the following examples: Murata, T., Fukami, K., & Fukagata, K. (2020). *Nonlinear mode decomposition with convolutional neural networks for fluid dynamics*. *Journal of Fluid Mechanics*, 882, A13; Akkari, N., Casenave, F., Hachem, E., & Ryckelynck, D. (2022).; Maulik, R., Lusch, B., & Balaprakash, P. (2021). *Reduced-order modelling of advection-dominated systems with recurrent neural networks and convolutional autoencoders*. *Physics of Fluids*, 33(3)). In our study, we took an approach based on splitting the temporal dynamics into two intervals: $[0, t]$ and $[t+1:t_{end}]$, so the β -VAE is trained using only information from the first part of the numerical simulation. We prove that the latent spaces we obtained are robust and are representative of the physics even in snapshots unrelated to the training split due to the chaotic nature of our example. The generalisation was achieved by choosing an appropriate model complexity that better balances the bias–variance tradeoff and by traditional hyperparameter tuning.
- Our study presents an entirely data-driven approach, i.e. that does not rely on auxiliary partial differential equations. Furthermore, our method has been tested in a complex flow, namely the 2D incompressible, two-dimensional, viscous flow around two collinear flat plates. This configuration leads to chaotic flow described by the Navier-Stokes equations, and constitutes a significant step forward with respect to the classic 2D low-Reynolds-number cylinder flow (which is not chaotic) or the Burgers equation, which have been used in most of the studies in the literature. Therefore, we argue that the flow we analyse is significantly complex, beyond previous works relying on simplified “toy problems”.
- In this study, modern transformer architectures have been tested, such as: the novel Easy Attention (see Sanchis-Agudo, M., Wang, Y., Duraisamy, K., & Vinuesa, R. (2023). *Easy attention: A simple self-attention mechanism for Transformers*. *arXiv preprint arXiv:2308.12874*). This method has been proved to be an improvement in comparison with the standard Self-Attention approach (which we used in the first version of the manuscript). Instead of relying on keys, queries and values to compute the attention score, the easy-attention mechanism directly treats the attention scores as learnable parameters, a fact that enables the model to capture the long-term temporal dynamics in chaotic systems.

Figure: Comparison of average prediction error over the prediction horizon in latent space, for self-Attention and easy-attention transformers.

- Moreover, we have tested several ROMs developed using different time steps. We have observed that the Transformer-based architectures are more efficient than the classical LSTM when the range of frequencies involved in the prediction is high. This can be explained by the ability of the transformers to learn phenomena associated with different time scales. These analyses are performed not in terms of reconstruction error, but also using Poincaré maps. This allows us to analyse the robustness of the state prediction, in order to show the potential of these approaches to work even as a potential state-estimator model, as reported with other approaches based on Kalman filters (see Colburn, C. H., Cessna, J. B., & Bewley, T. R. (2011). *State estimation in wall-bounded flow systems. Part 3. The ensemble Kalman filter. Journal of Fluid Mechanics*, 682, 289-303). These models are helpful to tackle the problem of controlling a turbulent flow based on its estimated state.

1. Eqs. (1) and (3): Instead of $\times 100$, $\times 100\%$ (or removing $\times 100$) will make more sense as 100% is unity. Similarly, “multiplied by 100” in the 3 lines below Eq. (1) looks weird and inconsistent with “A value of 1” below Eq. (2).

We apologise for the confusion. We agree with the reviewer that the nomenclature can lead to confusion. We have added the % as suggested since it is clear and more meaningful.

2. Flow configurations described on Page 3: It is hard to understand the flow configurations without figures, although I can roughly guess from Figs. 2, 4 etc.

We apologise for the confusion. We have included a schematic representation of the numerical setup to help the reader understand the flow configuration.

Figure. Schematic representation of the numerical setup.

3. Page 3: I am a bit confused by the meaning of the subscript i to \mathbf{s}_i , \mathbf{r}_i , and \mathbf{a}_i , because it is added to the bold-face vectors. Doesn't i represent the mode number? I cannot understand the meaning of “the desired i^{th} position” on the same page --- what does it mean by "position"? The same applies to \mathbf{r}_{i_j} on Page 5.

We thank the Reviewer for their observation. We have revised and renamed some variables to make the manuscript clearer:

\mathbf{r}_i , renamed to $\mathbf{r}_i(t)$, refers to the i^{th} time series of latent space variables.

$\mathbf{a}_i(t)$ refers to the i^{th} time-dependent POD coefficient series, as defined in the Methods section.

\mathbf{s} refers to a vector in the latent-space to be decoded, to analyse the latent space, for example.

\mathbf{s}_i refers to a one-hot encoded vector in the latent-space to be decoded, that selects the mode to be individually decoded.

4. Related to the comment 3 above, I am curious to see how the trajectory of the latent variables is for the periodic case.

We are happy to show the latent space of the periodic case and we have included the trajectory of the latent variables for the periodic case with their respective prediction in the answer to the following question.

5. Fig. 8(a): The instantaneous behaviors are not well reproduced by any models, which is somehow typical to nonlinear cases. But I am curious whether the present Transformer model works well for the periodic case. For instance, Hasegawa et al. (Fluid Dyn. Res. 2020) reports the LSTM-based ROM suffers from a substantial L2 error even in a periodic case due to a small phase error accumulating every step.

We did not include the figure due to the simplicity of the case, but here are the resulting latent variables for the periodic case. We have added them to the manuscript with the corresponding temporal predictions. In our case, at least for the available test data length

($100t_c$), we did not appreciate such an error in the transformer or the LSTM, so in our model, the phase shifting error could be lower than in previous studies. We would also like to remark that thanks to the inclusion of extra data and improvements in the neural network architectures the temporal predictions have been improved.

Figure. Test series prediction for periodic case.

6. Related to the comment 5 above, "The transformer produces accurate predictions" on Page 9 may better be expressed as "The transformer produces more accurate predictions".

We thank the reviewer for their suggestion. We have improved this section and reworked it to include even more transformer architectures.

7. Pages 12-13: From my own experience, the network structure and its depth used in the present study looks reasonable for the present flow problems. But can the authors note any comments when the flow becomes more complex, e.g. turbulent channel flows?

We thank the reviewer for their consideration. We have found that the model needs to be more complex to learn more complicated flow patterns. One of our findings is that the key point to obtain generalisation of latent spaces is not the number of snapshots but the number of dynamic flow phenomena that the neural network observes during its training. The required increase in both computational resources and available data to deal with the increased complexity of the flows drives the problem beyond the scope of current work. But we recognize this is an open question we want to work on.

8. Page 13: I am just curious why the authors chose ELU, although there are some previous studies (e.g., Fukami et al., SN Comput. Sci. 2021) which show that ELU (when used in CNN-AE) is relatively robust against different flow problems.

We thank the reviewer for their suggestion, and we have added this reference. ELU is chosen for two reasons: first, its derivative at the origin is continuous, which should better reproduce the continuous physical system under consideration. The second reason is that the derivative of the function is one for positive inputs, which mitigates the gradient-vanishing problem for deep networks, as shown in (D.-A. Clevert, T. Unterthiner, and S. Hochreiter, Fast and accurate deep network learning by exponential linear units (ELUS), arXiv preprint arXiv:1511.07289 (2015)).

9. Page 13, 2nd line from the bottom: I cannot understand "The last 1024 time steps are then used as test data" which, for the periodic case, looks to exceed "1,000 instantaneous flow fields" on Page 3.

We apologise for the confusion. For the periodic case, the train/test split values used are 900 and 100, respectively. We have corrected this issue in the manuscript.

Comments to Reviewer #3:

The authors would like to thank the Reviewer for their comments and thorough examination of our work. The Reviewer's comments are reproduced below, with our responses following each comment in blue text. The changes are highlighted in blue in the text. The paper has been revised, taking the outlined changes into account; thus, we are attaching a non-marked-up version to allow easier reading of the manuscript. Please note that the numbering follows the marked-up version when citing references and figures in this rebuttal.

Before proceeding with the rebuttal, we would like to remark that we have substantially reworked the discussion of the results following the referees' comments.

Reviewer #3 (Remarks to the Author):

The authors present a β -VAE framework to identify the modes of two relatively simple canonical test cases of flow dynamics. The authors then predict the evolution of the flow with a transformer architecture. I do not think that the paper in its current form can be published in this journal.

We understand the reviewer's comments regarding the lack of novelty due to the large number of recent studies on flow-field forecasting and potential applications to the fluid-dynamics field. However, we would like to highlight some key points regarding our study and its outcomes, which justify its uniqueness. Our final goal is not to develop a flow-field forecasting model such as in references provided by the reviewer, like <https://arxiv.org/abs/2211.02556>, but instead to develop a Reduced-Order model that allows us to reduce the dimensionality of the problem, enabling us to:

-Develop state-estimator models in a cheaper computational manner since developing forecasting approaches in a low-dimensional domain is faster and cheaper. These models can be very accurate in image reconstruction, but in the present context they must be able to predict the state of the flow to be used in frameworks of Active Flow Control, where real-time predictions need to be made.

-Develop secondary models where the information of the low-dimensional domain can be correlated with sensors which allow us to predict the flow state and even reconstruct the flow field using the decoder.

-Analysing the features and structures obtained in the low-dimensional manifold, the flow dynamics can be studied and analysed to uncover dominant phenomena.

To study the potential use of a Reduced-Order Model, it is relevant to assess not only its potential to perform flow-field reconstructions but also whether it can capture the dynamics of the systems, and this part has not been analyzed so extensively in the literature. In fact, many recent studies have been performed in analytical flows, such as the Burgers equation, non-chaotic regimes or 2D doubly-periodic domains. Before applying these techniques to complex flows, such as turbulent flows at high Reynolds numbers, it is necessary to assess their validity in numerical simulations of chaotic flows (such as the ones studied here) with a

component of geometrical complexity that allow us to gather a large number of snapshots while having a reasonable computational cost. Such flows still pose a big challenge, and constitute a necessary step before analysing high-Re turbulence.

With this in mind, we have improved the manuscript's quality and hope to address all the reviewer's concerns. In particular, we would like to highlight the following inputs:

- We have extended our numerical simulation data to 150,000 snapshots with an interval between snapshots of 0.2 convective times. This database has allowed us to perform a detailed analysis comparing the influence of the number of snapshots and dynamic cycles to generalise the latent space. In addition, this database has also allowed us to compare the performance of the different neural-network architectures in the time-series forecasting for different intervals between snapshots and different time-scale ranges.
- In most prior studies based on data-driven approaches, the splitting data procedure is performed using random splitting, and therefore, the obtained ROMs and metrics are biased by the fact that the model has observed information of the complete flow dynamics (see the following examples: Murata, T., Fukami, K., & Fukagata, K. (2020). *Nonlinear mode decomposition with convolutional neural networks for fluid dynamics*. *Journal of Fluid Mechanics*, 882, A13; Akkari, N., Casenave, F., Hachem, E., & Ryckelynck, D. (2022).; Maulik, R., Lusch, B., & Balaprakash, P. (2021). *Reduced-order modelling of advection-dominated systems with recurrent neural networks and convolutional autoencoders*. *Physics of Fluids*, 33(3)). In our study, we took an approach based on splitting the temporal dynamics into two intervals: $[0, t]$ and $[t+1:t_{end}]$, so the β -VAE is trained using only information from the first part of the numerical simulation. We prove that the latent spaces we obtained are robust and are representative of the physics even in snapshots unrelated to the training split due to the chaotic nature of our example. The generalisation was achieved by choosing an appropriate model complexity that better balances the bias–variance tradeoff and by traditional hyperparameter tuning.
- Our study presents an entirely data-driven approach, i.e. that does not rely on auxiliary partial differential equations. Furthermore, our method has been tested in a complex flow, namely the 2D incompressible, two-dimensional, viscous flow around two collinear flat plates. This configuration leads to chaotic flow described by the Navier-Stokes equations, and constitutes a significant step forward with respect to the classic 2D low-Reynolds-number cylinder flow (which is not chaotic) or the Burgers equation, which have been used in most of the studies in the literature. Therefore, we argue that the flow we analyse is significantly complex, beyond previous works relying on simplified “toy problems”.
- In this study, modern transformer architectures have been tested, such as: the novel Easy Attention (see Sanchis-Agudo, M., Wang, Y., Duraisamy, K., & Vinuesa, R. (2023). *Easy attention: A simple self-attention mechanism for Transformers*. *arXiv preprint arXiv:2308.12874*). This method has been proved to be an improvement in comparison with the standard Self-Attention approach (which we used in the first version of the manuscript). Instead of relying on keys, queries and values to compute the attention score, the easy-attention mechanism directly treats the attention scores as learnable parameters, a fact that enables the model to capture the long-term temporal dynamics in chaotic systems.

Figure: Comparison of average prediction error over the prediction horizon in latent space, for self-Attention and easy-attention transformers.

- Moreover, we have tested several ROMs developed using different time steps. We have observed that the Transformer-based architectures are more efficient than the classical LSTM when the range of frequencies involved in the prediction is high. This can be explained by the ability of the transformers to learn phenomena associated with different time scales. These analyses are performed not in terms of reconstruction error, but also using Poincaré maps. This allows us to analyse the robustness of the state prediction, in order to show the potential of these approaches to work even as a potential state-estimator model, as reported with other approaches based on Kalman filters (see Colburn, C. H., Cessna, J. B., & Bewley, T. R. (2011). *State estimation in wall-bounded flow systems. Part 3. The ensemble Kalman filter. Journal of Fluid Mechanics*, 682, 289-303). These models are helpful to tackle the problem of controlling a turbulent flow based on its estimated state.

Major general concerns:

1. Novelty: This is an interesting paper, but nothing substantially novel to demand a publication in Nat Comm. The authors used a well-established VAE architecture to identify a latent space that is argued to represent the true modes of the system. It is validated by a very simple periodic example that is not a turbulent flow. Then the authors use another well-established transformer architecture to predict the evolution of the modes. Again, there is no novelty there. Besides, the time series forecasts shown by the authors do not reflect how transformers are doing substantially better than the other baselines, which show pretty low-skill in themselves. Fig 8b shows transformer has a larger error than the other two baselines. Also, why does KNF start off with a bump in that figure?

We would like to first note that the test case under consideration does not exhibit periodic behaviour. It is a chaotic flow, with complex temporal dynamics, which goes far beyond other test cases in the literature. We believe that this case constitutes a significant step forward in complexity, posing a real challenge to the proposed architectures. Furthermore, we have worked on improving this section. As a result, we have included a new modern transformer architecture known as Easy Attention (see Sanchis-Agudo, M., Wang, Y., Duraisamy, K., &

Vinuesa, R. (2023). *Easy attention: A simple self-attention mechanism for Transformers*. *arXiv preprint arXiv:2308.12874.*), which has been proved to be an improvement in comparison with the standard Self-Attention approach (which we used in the first version of the manuscript). Instead of relying on keys, queries and values to compute the attention score, the easy-attention mechanism directly treats the attention scores as learnable parameters, a fact that enables the model to capture the long-term temporal dynamics in chaotic systems.

The KNF starts with a bump due to having a significant starting error from the $t+1$ instant. All the predictions in this figure are recursive, being the $t+2$ prediction obtained from the predicted $t+1$, and therefore, the error is increasing exponentially. It has to be remarked that the error is not calculated only based on the results of Figure 7 which is an example of a trajectory since it is an ensemble error computed over 100 evenly spaced windows in the test data to capture the average trend. The observed bump only reports that, on average, the initial error of the KNF over the 20 modes is consistently higher than the other models.

The details of all the new predictions and comparisons are provided below:

FIG. 7: Example of a trajectory of the latent-space modes with their associated predictions by different models: (a) $Re = 40$, $\alpha = 90^\circ$, (b) $Re = 100$, $\alpha = 80^\circ$.

2. Assumptions: There has been a lot of previous work in POD-based decomposition and then prediction; AE-based decomposition and then prediction, etc, etc. Studies looking at these canonical systems often ignore the fact the real turbulent flow at $Re > 10K$ or so has much more complexity. And more importantly, the relevant modes in those systems, arising due to the non-normality of the flow, are not orthogonal-- a central assumption in this paper, following many other studies. This has huge implications and has been studied in other fields of turbulence like geophysical flows; see

1. <https://rmets.onlinelibrary.wiley.com/doi/abs/10.1002/qj.305>
2. <https://journals.ametsoc.org/view/journals/atsc/73/9/jas-d-16-0099.1.xml> .

We thank the Reviewer for this important comment. We would like to note that, although the cases under study are not turbulent, they pose a significant challenge beyond results previously published in the literature, which are mostly based on periodic flows. The present study focuses on flows which exhibit chaotic behaviour. Although the flows studied here are not turbulent, they constitute a necessary step before tackling turbulent flows, and here we thoroughly assessed the proposed models to achieve the best possible performance.

On the other hand, we agree with the reviewer that non-normality is a very important feature of a variety of turbulent flows, across a range of Reynolds numbers. Note, however, that it is still possible to capture non-normal behaviour when using an orthogonal set of basis functions. For example, one can project the full dynamics (e.g. the Navier-Stokes equations) onto a subspace spanned by orthogonal modes, and still obtain a non-normal (linearized) operator (though for highly non-normal systems the best direction of projection is not necessarily orthogonal to this subspace). On a similar note, rather than projecting onto known governing equations, it would also be possible to obtain a data-driven model for the evolution of mode coefficients. Again, even if the modes themselves are orthogonal, the eigenvectors of the identified (linearized) system would not necessarily be orthogonal (and ideally would capture the nonnormality of the full system).

Simple examples and discussion of these ideas can be found, for example, in:

1. Ilak *et al.*, “Model Reduction of the Nonlinear Complex Ginzburg–Landau Equation,” 2010 (<https://epubs.siam.org/doi/pdf/10.1137/100787350>)

2. Rowley & Dawson, “Model Reduction for Flow Analysis and Control,” 2017 (<https://www.annualreviews.org/doi/10.1146/annurev-fluid-010816-060042>)

3. State-of-the-art (SOTA): Data-driven prediction of turbulent flow fields has seen a lot of progress in the geophysical flows community. For example, data-driven architectures can predict real weather dynamics (not canonical systems) at a much higher resolution (than what is studied in this paper) and outperform numerical simulations. These are real systems with SOTA archs achieving better performance. See 1. <https://arxiv.org/abs/2212.12794>; 2. <https://arxiv.org/abs/2211.02556>; 3. <https://arxiv.org/abs/2202.11214>. Even in canonical turbulent flow with high Re (both 5k and 10K), other authors have shown SOTA results, see 1. <https://arxiv.org/abs/2010.08895> 2. <https://openreview.net/forum?id=1C36tFZn7sR>

Looking at these results, it seems that one can actually predict turbulent flow (at a much higher Re compared to this study) without resorting to “just” predicting the temporal evolution of “modes”. This raises the question as to why we must even go to this route except for the fact that there are many other studies that do this using some flavor of AE to extract some modes and some flavor of a temporal integrator to predict the modes for simple systems.

I still believe there is a lot of value in fundamental studies looking at modal decompositions with incremental skill but improved interpretability, yet this particular paper does not have the novelty or address any fundamental theory to be published in Nat Comm. I feel it is more suitable elsewhere in a more technical journal.

We agree with the reviewer that data-driven weather prediction models can predict turbulent flows at a larger scale and resolution. However, this paper aims to decompose the flow field into a low-dimensional space that can be predicted in time with low computational cost or mapped into other low-dimensional spaces. Weather prediction models rely on a dense source of information that is only sometimes available in practical scenarios. The type of ROM developed in our work is more oriented towards applications such as Active Flow Control, where only a few sensors are usually used to estimate the system's state, and future trajectories must be propagated in real time. The traditional POD may be sufficient for simple cases (<https://arxiv.org/abs/1706.03531>), but the aim here is to extend this idea to

more complex flows using a non-linear model. Furthermore, the interpretability of the obtained modes (in a disentangled latent representation) enables studying additional aspects of the physics when analysed in the latent space. Consequently, in the context of ROM development from sparse measurements, both with a focus on flow control and optimization, the approach proposed here is novel and provides a significant step forward in terms of performance and model capabilities.

In this manuscript, we have attempted to compare our proposed methodology with several other related data-driven methods to benchmark its performance, for the modal decomposition part we used the Proper Orthogonal Decomposition (POD) but now we also include the vanilla Autoencoder (AE) and classical Variational Autoencoder (VAE) to justify the potential of the β -VAE approach. Being the results summarised in the following tables, where data with different temporal resolutions ($1t_c$ and $0.2t_c$ between snapshots) have been used:

Method	E Train $1t_c$	E Test $1t_c$	E Train $0.2t_c$	E Test $0.2t_c$
β -VAE 20-modes	90%	80%	91%	76%
β -VAE 10-modes	82%	62%	86%	58%
POD 10-modes	45%		45%	
POD 20-modes	63%		63%	

Method	MSE Train	MSE Test
β -VAE 20-modes ($\beta=0.05$) ($1t_c$)	0.0716	0.177
β -VAE 20-modes ($\beta=0$)	0.0387	0.2159
VAE 20-modes ($\beta=1$)	0.2612	0.3305
AE 20-modes	0.0408	0.2345

For the part of the time series forecasting, we have included a new Transformer architecture, easy attention, and we have included a comparison with the classical Transformer architecture based on self-attention, with an LSTM and a KNF model.

FIG. 8: Analysis of the predictions in latent-space for the case with $Re = 100$, $\alpha = 80^\circ$: (top) Average prediction error over the prediction horizon, (bottom) Poincaré sections with plane $r_1 = 0$ and $dr_1/dt > 0$, for case with latent-space size $d = 20$ and $\Delta t = t_c$. Black lines correspond to the original data and blue lines to the predicted data.

Having said the above, we acknowledge that In the future, additional comparisons to a broader class of methods (such as those suggested by the reviewer), and across a broader range of test problems, could help further contextualise and benchmark our methods.

Major Technical Comments:

1. Figure 5 shows 10 modes. What is the true mode corresponding to these predictions? What is the reason to believe that they are close to the true mode? In this simple example, I understand that the authors have a general sense of how a mode might look like, since it differs ever-so-slightly from the periodic case, but how does the VAE framework help us study a more general setting with high Re ? Is this framework generalizable in that sense?

We thank the Reviewer for this comment. There is not any “true mode”, since that depends on the particular modal decomposition employed to study the flow. We refer to the POD modes because they constitute a widely established framework, including the interesting properties of optimality and orthogonality. However, these modes rely on a linear reconstruction, thus not being very efficient in compressing the original high-dimensional data. The modes we obtain from the β -VAE are associated with a non-linear reconstruction, which means that the leading modes contain the dominant physics plus additional non-linear fluctuations which would be found deep into the POD spectrum when performing a linear modal decomposition. This was discussed in our previous study by Eivazi et al. (see Eivazi, H., Le Clainche, S., Hoyas, S., & Vinuesa, R. (2022). *Towards extraction of orthogonal and*

parsimonious non-linear modes from turbulent flows. Expert Systems with Applications, 202, 117038.), where we compared POD and β -VAE modes and found these interesting connections. Also note that, in this figure, the first modes have a larger contribution to the reconstruction and are therefore associated with more dominant physical phenomena in the flow.

2. Fig 8 (a) and (b). From (a) I feel that all the methods are doing as well or as poorly. Then why is there an emphasis on the transformer? From (b), it feels that the transformer is doing worse(?) Also, why does the KNF have a big jump at the first time step in Fig 8(b)?

This point raised by the reviewer is important and deserves clarification. We focus on the transformer because, as shown in Figure 9(c), its Poincaré map is in good agreement with that of the reference, whereas the maps from the LSTM and the KNF are significantly worse. We would like to note that in Figure 8 we do not expect the models to obtain a perfect agreement with the reference at all times. This is because the predictions into the future rely on predicted data, thus leading to an exponential increase of errors. Thus, even if a very small error is made in the predictions, the chaotic nature of the system will lead to a different trajectory, even if the correct physics have been learned by the neural network. In other words, we learn the correct physics up to a very small error (as shown in Figure 9), which means that we learned the dynamics of the system, and Figure 8 just shows that we converge to a different trajectory due to the small accumulation of errors. This is reflected in Figure 8(b), which exhibits such an error accumulation in all the cases. This is discussed in detail in our previous work by Srinivasan et al. (see Srinivasan, P. A., Guastoni, L., Azizpour, H., Schlatter, P. H. I. L. I. P. P., & Vinuesa, R. (2019). Predictions of turbulent shear flows using deep neural networks. *Physical Review Fluids*, 4(5), 054603.) This is further clarified in the manuscript as follows:

As seen in Figure 10(a), all predictions will diverge over time as small initial fluctuations in the model accumulate over time, to analyse if the predicted velocity fields still reproduce the dynamics of the system, we compare the POD of the original test data with the predicted fields for the same data, in Figure 10(b) the results are compared, showing that the predicted fields still produce similar POD modes as the original data. These results reinforce the quality of the reconstruction observed in Figure 9 and show that the model adequately reproduces the flow dynamics, reinforcing the robustness of the methodology proposed in this work.

As stated in the previous answer, the KNF starts with a bump due to having a significant starting error from the $t+1$ instant. All the predictions in this figure are recursive, being the $t+2$ prediction obtained from the predicted $t+1$, and therefore, the error is increasing exponentially. It has to be remarked that the error is not calculated only based on the results of Figure 7 which is an example of a trajectory since it is an ensemble error computed over 100 evenly spaced windows in the test data to capture the average trend. The observed bump only reports that, on average, the initial error of the KNF over the 20 modes is consistently higher than the other models.

3. Fig 9 is actually really good. It shows the advantage of having a transformer. But still, the Poincare section does not correspond to the truth although it spans a larger space of the truth. If I were to guess, all three methods are suffering from the fact the β -VAE modes are not really representing the most relevant dynamical modes. Also, what is the rationale

behind them being the most physical modes? The fact that the authors made them orthogonal? But that's just as bad as POD modes-- isn't it?

This comment is related to the two previous ones. The point of making the modes orthogonal is to better represent the physics of a flow that is nonlinear, and therefore a β -VAE with orthogonal modes would better capture the dominant physics in such a nonlinear scenario than POD. Even if the Poincaré map is not perfect, it is indeed in very good agreement with the reference, and certainly outperforms other similar results in the literature. Since the modes are ordered in terms of their contribution to the reconstruction, we argue that the first modes are associated with more relevant flow dynamics. As mentioned in a comment above, this was discussed by Eivazi et al. (see *Eivazi, H., Le Clainche, S., Hoyas, S., & Vinuesa, R. (2022). Towards extraction of orthogonal and parsimonious non-linear modes from turbulent flows. Expert Systems with Applications, 202, 117038.*), when comparing betaVAE and POD modes.

4. Does the long predicted trajectory have the same POD modes as the truth? It's a simple test to study first-order variability? How about the PDF?

We thank the reviewer for their fruitful suggestion. We have calculated the POD and the PDF of the latent variables from the predicted and the true test snapshot data of the Easy Attention + β -VAE model in the following figures, included in the manuscript:

Figure. Comparison of the four most energetic POD modes of test data and POD modes of encoded, predicted and decoded test data. Easy-attention transformer with $d=20$ case. (left) u modes, (centre) v modes, (right) spectra comparison.

Figure. Comparison of the Probability Density Functions of the first 6 temporal modes for the test data.

It can be observed that the resulting spatial modes POD for both predicted and true data have similar features and characteristic frequencies, as the predicted fields appear to capture the large scale phenomena observed in the POD modes from the test data. In the PDF analysis of the latent space variables, we observe in a clear way that the easy

attention model appears to be the best model in capturing this behaviour of the latent variables, being its distribution closest to the reference and without having any peak or anomaly compared to the LSTM or the KNF which appear to fail more in the extreme value distributions.

5. There is a lot of emphasis on visualization in the paper, but there is not enough conviction that the modes visualized are the right modes if we don't know the true modes of the system (which is obviously close to impossible for a real system, but even for this one?)

In relation to Comment 1, it is not clear what is meant with “true” modes, as these will depend on the employed decomposition. We believe that the β -VAE provides the best of two worlds, namely bringing the orthogonality and optimality of POD and combining them with the non-linearity which would be able to represent non-linear effects in the flow. Therefore, and as we show in the paper, the identified modes can provide a compact and effective reduced-order representation of the full system.

6. Finally, I think the system chosen is too canonical and has little similarity to large-scale practical systems and hence many of the arguments might be too fine-tuned to this toy problem.’

As mentioned a couple of times above, this flow case is not a “toy problem”, as it exhibits chaotic behaviour, and it constitutes a necessary step in complexity between the existing studies on periodic flow and complete turbulent flows. In the revised version of the manuscript we developed a framework which performs very well on data not used to build the reduced-order model (as opposed to what is done in the literature), a fact that indicates that the model is quite general and our results are not too fine tuned to this case or the employed data. Finally, the provided examples on geophysical flows are interesting, but are not really what we are aiming at here, as we want to use these ROMs for flow-control applications in high-Reynolds-number turbulent flows from sparse measurements.

Despite the fact that our Reynolds number is not high, we would like to note that the domain is much more complex than a 2D doubly-periodic domain. Therefore, we are adding a component of geometrical complexity, which is a necessary step to adopt these methodologies to industrial flows.

We want to thank the Reviewer again for all the time spent in the review, and the useful suggestions which have certainly improved the quality and clarity of the manuscript. We hope that we could show the Reviewer the interest of this work for the turbulence community and that the Reviewer will see the value of this contribution in that context.

Reviewers' Comments:

Reviewer #1:

Remarks to the Author:

Thank you for the improvement of the paper.

Reviewer #2:

Remarks to the Author:

All the concerns I had on the previous manuscript (although minor) have properly been reflected in the revised manuscript.

I am happy to recommend this version of manuscript for publication in Nat. Commun.

Reviewer #3:

None